# Physico-chemical aspects of Thai fermented fish viscera, *Tai-Pla*, curry powder processed by hot air drying and hybrid microwave-infrared drying

Warongporn Choopan[1], Worawan Panpipat[1]*, Mudtorlep Nisoa[2], Ling-Zhi Cheong[3], Manat Chaijan[1]

1 Food Technology and Innovation Research Center of Excellence, School of Agricultural Technology and Food Industry, Walailak University, Nakhon Si Thammarat, Thailand, 2 School of Science, Walailak University, Nakhon Si Thammarat, Thailand, 3 Department of Food Science and Engineering, School of Marine Science, Ningbo University, Ningbo, China

* pworawan@wu.ac.th

**Data Availability Statement:** All relevant data are within the manuscript.

## Abstract

The objective of this research was to comparatively investigate the effect of hot air drying (HA) and hybrid microwave-infrared drying (MI) on physico-chemical characteristics of Thai fermented fish viscera, *Tai-Pla*, curry powder (TCP). HA was carried out at 60°C, 70°C, and 80°C and MI was carried out at a microwave power of 740, 780, and 810 W with a constant infrared heating power (500 W) for different drying times to obtain the final moisture content $\leq 12.0\%$ and the water activity ($a_w$) $\leq 0.6$. The quality characteristics of TCP were governed by HA temperature and MI output power. TCP dried using HA and MI at all conditions had similar contents of protein, lipid, ash, fiber, and carbohydrate (p>0.05). The fastest drying rate was detected when MI at 810 W for 40 min was applied (p<0.05). In this condition, TCP had the lowest browning index ($A_{294}$ and $A_{420}$) and the highest lightness ($L^*$ value) (p<0.05). TCP dried with MI at all powers had higher phenolic content and lower TBARS compared to HA (p<0.05). However, no significant differences in DPPH$^\bullet$ scavenging activity were observed among TPC made by HA and MI (p>0.05). Similar Fourier transform infrared (FTIR) spectra with different peak intensities were observed in all samples, indicating the same functional groups with different contents were found. The bulk density of all TCP ranged from 0.51 g/mL to 0.61 g/mL and the wettability ranged from 24.02% to 26.70%. MI at 810 W for 40 min effectively reduced the drying time (5-fold faster) and lowered the specific energy consumption (18-fold lower) compared to the HA at 60°C for 210 min. Therefore, MI is a promising drying technique to reduce the drying time and improve the overall quality of TCP.

## Introduction

Thailand has long been recognized as "the kitchen of the world" with varieties of tastes and flavors. Strong aromatic components and a spicy edge of Thai cuisine are caused by the

**Funding:** This research was financially supported by Research and Researchers for Industries (RRI) program and Shaw Processing Food Co. Ltd. [Grant No. MSD61I0053] and the new strategic research project (P2P), Walailak University, Thailand. The funders had no role in study design, data collection and analysis, decision to publish, or preparation of the manuscript.

**Competing interests:** We received the co-funding grant from Research and Researchers for Industries (RRI) program and Shaw Processing Food Co. Ltd. [Grant No. MSD61I0053]. The main funding came from the RRI program and a partial funding from a commercial source: Shaw Processing Food Co. Ltd. We confirmed that this does not alter our adherence to all PLOS ONE policies on sharing data and materials.

formulation of spices, seasoning, and curry paste. Typically, the specific ingredients particularly spices and curry paste with medicinal benefits have long been used for creating the unique flavor in Thai food [1]. A curry of Southern Thai cuisine is one of the most favorite Thai food having a strong flavor and aroma, depending on the type of curry paste used [1,2]. Hot and spicy *Tai-Pla* curry is well recognized as a unique Southern Thai curry. Its name is derived from the key ingredient, *Tai-Pla*, which is a salty sauce made from fermented fish viscera [3]. This curry is usually served with fresh vegetables in a separate plate and eaten along with Thai rice noodle. *Tai-Pla* curry paste can be prepared by mixing herbal ingredients (e.g., dried chilli pepper, garlic, shallot, lemongrass, and galangal) with *Tai-Pla* sauce. Like other curry pastes, the *Tai-Pla* curry paste is typically perishable because of high moisture content [4]. In order to commercially distribute this unique local culinary treat over Thailand and export markets, the production of shelf-stable instant *Tai-Pla* curry powder (TCP) should be developed using an appropriate drying technology.

Drying is one of the most practical preservative methods used in the food sectors to reduce the amount of water from the raw materials and consequently improve the storage stability of food products [5]. The perishable fresh produces and high moisture minimal-processed foods are frequently subjected to drying to minimize the chemical and microbiological deteriorations [6]. There are various commercial drying techniques depending on the requirement of users and the target characteristics of the final product. Hot air convective drying (HA) is one of the most popular and cost-effective methods for agro-food products [5]. However, the major drawbacks of the HA are high drying temperature and long period of drying time, which can substantially result in the deterioration of product quality, such as color, flavor, texture, and nutrition [5,7]. Therefore, a promising drying technology has been developed such as microwave drying (MW) to improve the final product qualities and reduce the drying time. MW is based on dielectric heating by electromagnetic waves and it has several advantages [5]. Demiray et al. [8] indicated that the lower drying temperature, higher drying rate, homogeneous energy diffusion through the material, better space utilization, formation of suitable final product characteristics, and giving better process control are the advantages of MW. However, the main obstacle of MW is the discontinuous heat generation in the food product, which is highly affected by the condition of MW [9]. Also, the other well-known problem of MW is moisture accumulation at the food surface [10]. To encounter these problems, another heating source particularly infrared radiation (IR) is equipped with microwave drier to distribute more uniform heat. Hybrid microwave-infrared drying (MI) combined the time saving advantage of MW with surface moisture removal advantage of IR [10]. IR is the part of the electromagnetic spectrum that is predominantly responsible for the heating effect of the sun. IR energy is absorbed by the food surface and converted to heat [6]. Due to the short time and effective thermal processing, MI can be used to prevent the quality degradation of dried food products [6,11–13].

This present study demonstrated the potential applicability of an alternative MI to produce TCP compared to the traditional HA. The drying rate was monitored to clearly define the drying efficiency. The final product qualities in terms of physico-chemical characteristics were comprehensively determined. These results can be practically presented a promising drying method to prepare TCP with superior quality for further commercial application.

## Materials and methods

### *Tai-Pla* curry paste preparation

The main ingredients for *Tai-Pla* curry paste, including dried whole red chili pepper (5.6%), dried whole garlic (7.5%), fresh lemongrass (11.3%), fresh galangal (3.8%), fresh turmeric

(3.8%), fresh shallot (3.8%), dried black pepper (5.6%), dried seedless tamarind (7.5%), Thai fermented shrimp paste (1.9%), fresh kaffir lime leaves (1.9%), and *Tai-Pla* sauce (47.3%) were purchased from Nap Anuson Food Market, a local market in Thasala, Nakhon Si Thammarat, Thailand (8.6650˚ N, 99.9225˚ E). All the ingredients were produced locally in Nakhon Si Thammarat, Thailand. The fresh plant raw materials were hand-trimmed, washed with tap water, and drained with a nylon screen for 30 min at room temperature (27–30˚C). Thereafter, all ingredients with the specified proportion were mixed and coarsely ground for 10 min using a grinder (MK 5087M Panasonic Food Processor, Selangor Darul Ehsan, Malaysia) to obtain fresh curry paste. The fresh curry paste (1,000 g) was then pasteurized at 90˚C for 10 min [14] in a controlled temperature Hanabishi HGP160S electric pan (Hanabishi Electric Co., Ltd., Bangkok Thailand) under continuous stirring. After cooling down to room temperature, the pasteurized curry paste with the moisture content of 64.7%, referred to as "fresh sample" was subjected to the drying experiment.

## Drying experiment

Two drying techniques including HA and MI were used to prepare TCP. The fresh samples (1,000 g) were uniformly spread in the tray with the thickness of 0.5 cm and subjected to dry using a traditional HA drier or an MI drier. The HA drier was operated in a DT 20S tray drier (4,500 W, Owner Foods Machinery Co., Ltd., Bangkok, Thailand) at 60˚C, 70˚C, and 80˚C with the circulation speed of 1 m/s. It has been reported that the maximum temperature used for drying the whole stink bean (*Parkia speciosa*) seed was at 70˚C using the same HA (tray drier) [5]. So, in this study, the drying temperatures were varied on the basis of 70±10˚C which were 60˚C, 70˚C, and 80˚C for TCP. The MI drier used in this study was developed by the Center of Excellence in Plasma Science and Electromagnetic Waves, Walailak University (Fig 1). The microwave output powers were adjusted to 740 W, 780 W, and 810 W with a fixed power of infrared heating (500 W) in order to meet the drying temperature of 60˚C, 70˚C, and 80˚C as done by a traditional HA drier. In Thailand, the water activity ($a_w$) of <0.65 and the moisture content of <13% (w/w) are the standards set for seasoning powder [15]. To comply with this standard and to ensure the food safety, the drying was proceeded until the final moisture content was reached ≤12% with the $a_w$ of ≤0.6. The drying curves between moisture content and drying time were plotted. The specific energy consumptions of HA and MI were calculated on the basis of the ratio of energy consumed to the initial mass of the samples and expressed as kJ/g [16]. To obtain the TCP, the dried samples were ground using a grinder for 2 min and passed through 35 mesh-sieve. The obtained TCP was packed in an aluminum foil laminated bag (10 cm × 15 cm) to prevent moisture adsorption and kept in an auto desiccator at room temperature for 24 h. Thereafter, the quality characteristics of the samples were determined.

## Determination of chemical characteristics

**Proximate composition and $a_w$.** The standard AOAC's methods [17] were used for proximate composition analysis including moisture (AOAC method number 950.46), crude protein (AOAC method number 928.08), fat (AOAC method number 963.15), ash (AOAC method number 920.153), fiber (AOAC method number 962.09), and carbohydrate. Carbohydrate content was estimated by difference (1).

$$\text{Carbohydrate content (\%)} = 100 - [\text{moisture} + \text{protein} + \text{fat} + \text{ash} + \text{fiber}] \qquad (1)$$

The $a_w$ was determined at room temperature using an Aqualab Series 3TE $a_w$ meter (Decagon, Pullman, WA, USA).

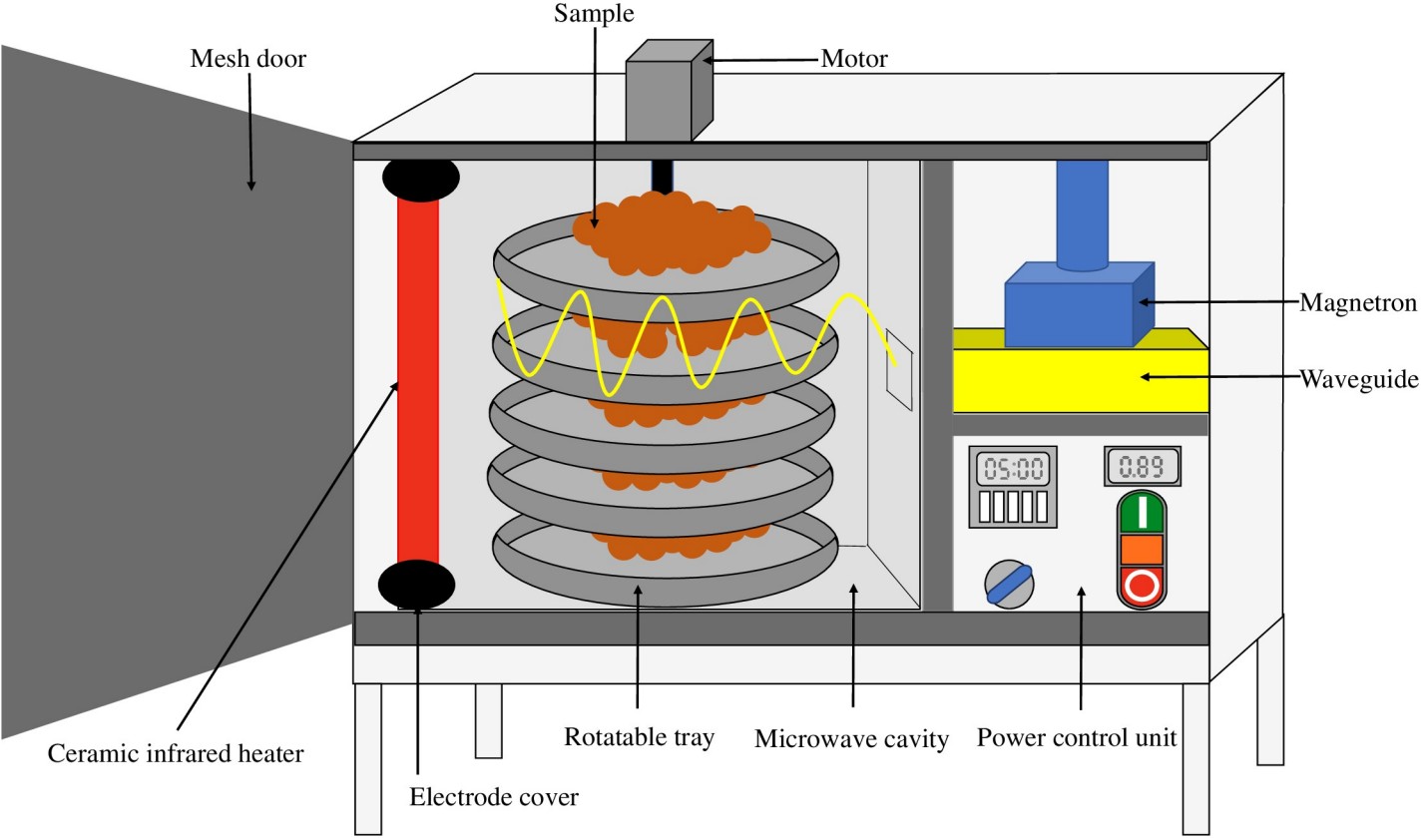

**Fig 1. Hybrid microwave-infrared drier.**

**Total phenolic content (TPC) and DPPH radical scavenging activity.** The samples (10 g) were extracted with 80% ethanol (10 mL) at 45°C in an incubator for 2 h with constant stirring. After filtration (Whatman No.1), the filtrates were analyzed for TPC using the Folin-Ciocalteu method [18]. The extracts (0.5 mL) were mixed thoroughly with 9.5 mL of distilled water, 1.5 mL of Folin-Ciocalteu reagent, and 1.2 mL of 7.5% sodium carbonate solution. After standing for 30 min at room temperature, the absorbance was determined at 765 nm (UV-Vis spectrophotometer, Shimadzu Scientific Instruments Inc., Columbia, MD, USA). A standard curve was prepared using gallic acid in the range of 0–100 ppm. The data were expressed as mg gallic acid equivalent (GE)/100 g.

DPPH radical scavenging activity was determined according to the method of Yen and Hsieh [19]. The ethanolic extracts (0.3 mL) was mixed with 1.5 mL of 0.10 mM DPPH in methanol solution and incubated at room temperature for 30 min. The absorbance was then read at 517 nm. The blank was prepared as the same manner, except that distilled water was used instead of the sample extract. A standard curve was prepared using Trolox in the range of 0–0.2 mM. The DPPH radical scavenging activity was expressed as mmoLTrolox equivalent (TE)/g sample.

**Thiobarbituric acid reactive substances (TBARS).** TBARS analysis is the most widely used method to determine the secondary lipid oxidation products [20]. Due to the presence of some lipid content in the *Tai-Pla* curry paste, the lipid oxidation can be taken placed during drying. Herein, the lipid oxidation was monitored using the TBARS. Samples (10 g) were mixed with 50 mL of distilled water and 2.5 mL of 4 M HCl and diluted to 100 mL with distilled water. The samples were distilled at 100°C following the procedure described by

Tarladgis et al. [21]. Then, 2 mL of distillate was mixed with 3 mL of a mixed TBA solution containing 0.375% thiobarbituric acid, 15% trichloroacetic acid, and 0.25 M HCl. A reagent blank was prepared by adding 2 mL of distilled water and 3 mL of mixed TBA solution. The mixture was heated in boiling water (100°C) for 10 min, followed by cooling with the running tap water. The absorbance was measured at 532 nm. A standard curve was prepared using malondialdehyde in the range of 0–10 ppm. The results were expressed as mg molondialdehyde equivalent/kg sample.

**Fourier transform infrared (FTIR) spectroscopy.**　The FTIR spectroscopy is a vibrational spectroscopic technique that can be used to characterize the substances by identifying their functional groups presented [22]. FTIR spectra (400–4000 $cm^{-1}$ with the resolution of 4 $cm^{-1}$ at the average of 16 scans) of the TCP were obtained using a horizontal Attenuated Total Reflectance (ATR) Trough plate crystal cell (45° ZnSe; 80 mm long, 10 mm wide and 4 mm thick) (Pike Technology, Inc., Madison, WI, USA) equipped with a Bruker Model Vector 33 FTIR spectrometer (Bruker Co., Ettlingen, Germany) at room temperature. Analysis of spectral data was carried out using the OPUS 3.0 data collection software program.

## Determination of physical characteristics

**Color.**　Colorimetric values of the samples were measured using a Hunterlab colorimeter (Hunter Assoc. Laboratory; VA, USA) with 10 standard observers and illuminant D65. The instrument was calibrated to a white and black standard. The $L^*$ (lightness), $a^*$ (redness/greenness), and $b^*$ (yellowness/blueness) values were recorded.

**Browning intensity.**　The browning intensity can be used to monitor the formations of the intermediate and final products of the Maillard reaction in the TCP. The presence of the Maillard reaction products (MRPs) affected both color and antioxidant activity of the food products [23]. The UV absorbance at 294 nm ($A_{294}$) was often used to indicate the intermediate MRPs while the final MRPs was monitored by the absorbance at 420 nm ($A_{420}$). An aqueous extraction and appropriate dilution were made prior to analysis. The mixture (1 g of TCP and 10 mL of distilled water) was homogenized at 13,000 rpm for 1 min at room temperature (IKA® Model T25 digital Ultra-Turrax®, Staufen, Germany). The centrifugation was applied at 3,000 ×g for 20 min at room temperature using an RC-5B plus centrifuge (Sorvall, Norwalk, CT, USA). Then, the absorbance of the supernatant was read measured at 294 nm and 420 nm using a Shimadzu UV-2100 spectrophotometer (Shimadzu Scientific Instruments Inc., Columbia, MD, USA) [23,24].

**Bulk density.**　The method of Jinapong et al. [25] was used for bulk density determination. The sample (1 g) was gently loaded into a 10 mL graduated cylinder and tapped for 10 times with the same strength. The volume was read directly from the cylinder and then used to calculate the bulk density according to the ratio of mass (g) to volume (mL).

**Wettability.**　To ensure the proper rehydration of the dried power, the wettability of all powders were determined [26,27]. Specifically, 40 mL of distilled water was transferred into a 250 mL beaker. Four grams of samples were added into the beaker and then stirred at 800 rpm for 5 min at room temperature. The mixture was centrifuged at 4,500 ×g for 5 min to separate the insoluble substances. Then, 20 mL of supernatant was transferred into a plate and dried at 100°C for 24 h. The wettability (%) was calculated as the weight of dry matters in the supernatants versus the weight of dry matters in the powders.

## Statistical analysis

A completely randomized design was used in this study. The data were expressed as means ± standard deviations (SD) of three replications (n = 3) for all analyses. Data analysis

was carried out using one-way ANOVA. The comparison of means was performed by Duncan's multiple-range test to identify significant differences (p<0.05) among samples [28], using the SPSS program (SPSS Inc., Chicago, IL, USA).

## Results and discussion

### Drying curves and specific energy consumption

The drying curves for TCP obtained from HA and MI are depicted in Fig 2A and 2B, respectively. The drying time was dependent on the drying technique and drying condition. HA at 60˚C showed the longest drying duration (210 min, $k \sim 0.2311$) (Fig 2A), whereas MI at 810 W showed the fastest drying rate (40 min, Fig 2B). Using MI, the faster moisture transfer from the interior of the samples to its surface was facilitated by MW [29] and the surface moisture removal was enhanced by IR [10]. TCP dried with HA at 70˚C and 80˚C had the drying time of 120 and 100 min, respectively (Fig 2A), which was shorter than that did at 60˚C for 1.75 and 2.10 folds, respectively. This was probably due to an acceleration of the moisture migration at high temperature. Similar result was attained for vacuum hot air dried Kumquat [9], in which a higher temperature can shorten a drying time. However, HA at 70˚C and 80˚C showed a negligible different in drying rate ($k \sim 0.3436$ vs $0.3907$) (Fig 2A). There were some differences in drying rate at the first 60 min (constant rate period) between 70˚C and 80˚C, then a slower water removal period was taken place (falling rate period). For MI, the higher MI output power rendered the shorter drying time (740 W for 70 min, 780 W for 45 min, and 810 W for 40 min), with the drying rate ($k$) of 0.7073, 1.0358, and 1.1446 for 740 W, 780 W, and 810 W, respectively. The higher the MI output, the faster the water removal [30]. Michalska et al. [31] indicated that microwave vacuum drying showed the lowest drying times for plum powder compared to freeze drying, vacuum drying, and convective hot air drying. In addition, Koç [32] reported that drying at high microwave output power resulted in an increase in the driving force of mass transfer and accelerating the rate of water vapor diffusion.

The specific energy consumptions of HA and MI are shown in Fig 2C. The HA consumed more energy than MI did. The highest specific energy consumption (57 kJ/g) was found in HA at the lowest temperature (60˚C) whereas the lowest specific energy consumption (3 kJ/g) was found in MI at the highest microwave power (810 W). From the results, the specific energy consumption was reduced as hot air temperature and microwave power increased in HA and MI, respectively. Chua et al. [16] reported that high drying intensities corresponded to a short drying duration, thereby reducing the specific energy consumption. In the MI, the MW heating helps to reach the temperature of the product quickly and the IR directly transfers the heat from IR emitters to the product surface without the need for any physical environment. Thus, the production of high-quality food with minimal energy consumption can be obtained [7].

### Chemical characteristics

**Proximate composition and $a_w$.** The moisture, protein, fat, ash, fiber, and carbohydrate contents of TCP dried using different drying methods with various drying conditions are presented in Table 1. The significant decrease in moisture content was observed in TCP (~10%) compared to the original fresh sample (~65%). This implied the shelf-stable of TCP by limiting the microbial growth during storage. No significant differences in moisture content among TCP prepared using different drying conditions were found (p>0.05), which was due to the clearly defined final moisture content. As expected, other proximate compositions, including protein, fat, ash, fiber, and carbohydrate of all TCP were gradually increased after removing significant amount of moisture compared to the original fresh paste (p<0.05). Morris and Barnett [33] suggested that the removal of water from the food matrix can increase the nutrient

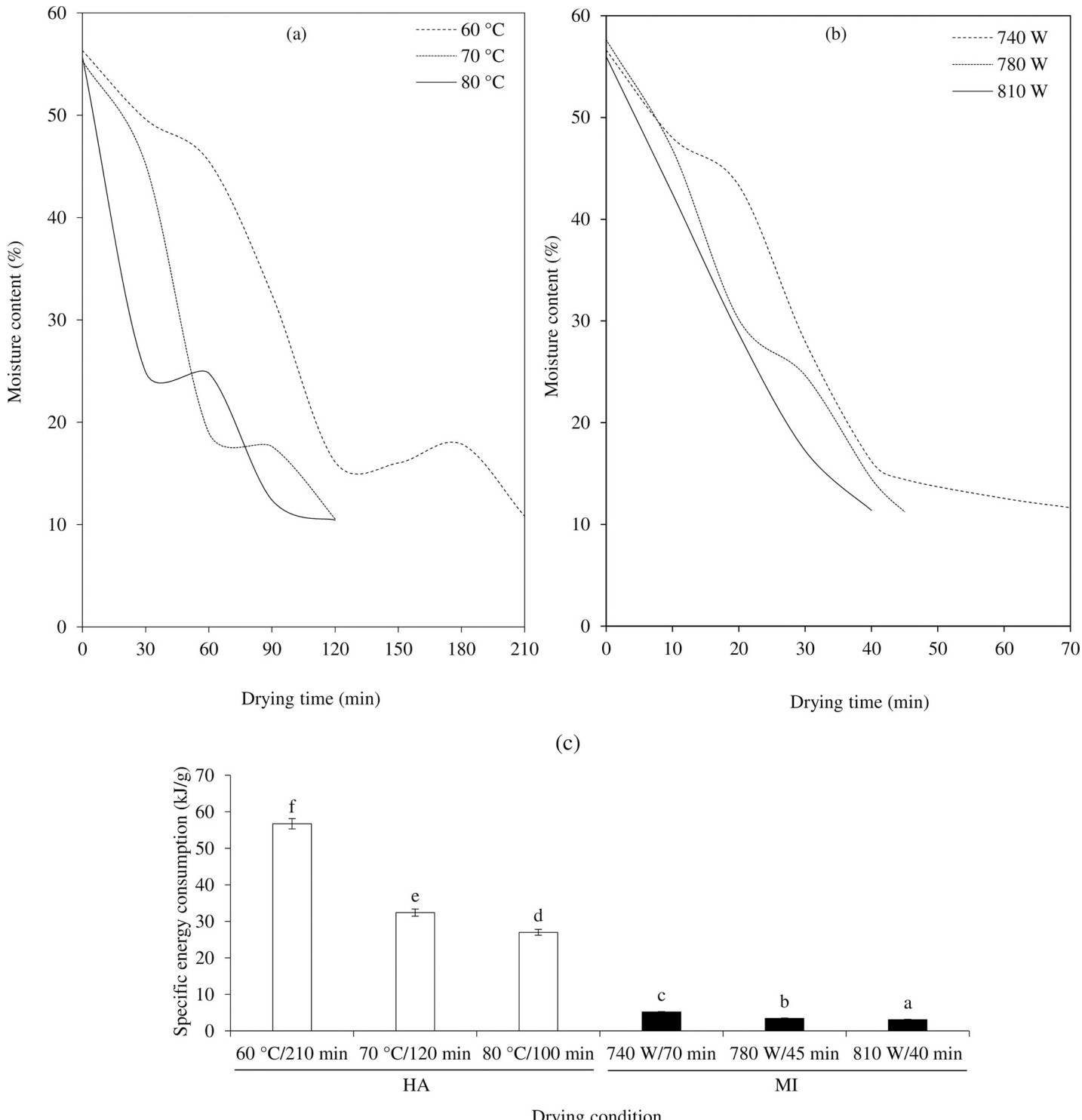

**Fig 2.** Drying curves of *Tai-Pla* curry powder (TCP) dried using hot air (HA) (a) and hybrid microwave-infrared drying (MI) (b) under different drying conditions.

concentration in dried food samples. Generally, no significant differences in protein, fat, ash, fiber, and carbohydrate were found among TCP dried using HA and MI (p>0.05). The crude protein contents of TCP ranged from 18.3% to 19.2% (p>0.05). These values were greater than

those reported for three commercial Nigerian seasoning powders (9.2–12.9%) by Lillian et al. [34]. This clearly caused by the addition of *Tai-Pla* (fermented fish viscera) sauce and fermented shrimp paste, as sources of protein, in the formulation of *Tai-Pla* curry paste. The crude fat contents of TCP ranged from 8.6% to 9.9% ($p > 0.05$), which were greater than the contents found in dried spices (e.g. *Tetrapleura tetraptera* [35] and *Scorodophleus zenkeri* [36]) and commercial Nigerian seasoning powders [34]. The presence of fat in the product may cause the oxidative instability during storage. The ash, crude fiber, and carbohydrate contents of all TCP were in the ranges of 18.3–19.7%, 13.6–14.9%, and 27.2–29.8%, respectively, indicating high levels of minerals, dietary fiber, and carbohydrate. These components originated from the ingredients used for preparation of fresh *Tai-Pla* curry paste. From the results, TCP can be classified as a nutritive flavoring agent. Summarily, the drying method and drying condition showed a negligible impact on proximate composition of the final TCP. This finding was in line with the study of Ojo et al. [37] who found no significant difference in protein content among sun-dried and solar-dried fufu flours. Moreover, an increasing in some food constituents after drying, particularly ash content, may contribute to the low volatility of certain substances e.g. minerals, which are slightly thermally degraded.

The $a_w$ of the fresh sample was 0.80 whereas the $a_w$ of the TCP was 0.44–0.53 (Table 1). The $a_w$ of all TCP was not significantly different ($p > 0.05$). Dried foods should have an $a_w < 0.60$ [38]. An $a_w$ of $<0.65$ and a moisture content of $<13\%$ (w/w) are the standards set for seasoning powder in Thailand [15]. It has been reported that the volumetric heating plus surface heating of MI can be effective for the microbial inactivation and restriction of moisture deteriorative physicochemical reactions [6]. So, the TCP can be considered as a shelf-stable product due to low moisture content and $a_w$.

*TPC.* A slight increment in TPC was observed in TCP compared to the fresh sample (Fig 3A). This was probably due to the evaporation of moisture, leading to the increased TPC concentration in the dried samples. Szychowski et al. [39] suggested that freeze-, convective hot air-, and vacuum- microwave driers increased the phenolic compounds in dried quince fruits compared to the fresh counterpart. Moreover, higher phenolic compounds in dried guava powders compared to fresh sample could be attributed to the concentration of the phenolic compounds after water evaporation [40]. The increment of phenolic compounds after drying could involve in the cell wall breakdown and disruption by heating effects and consequently eases the release and extractability of bound phenolic compounds [39]. Generally, plant

**Table 1. Proximate composition and water activity ($a_w$) of *Tai-Pla* curry paste (fresh sample) and *Tai-Pla* curry powder (TCP) dried using hot air (HA) and hybrid microwave-infrared drying (MI) under different drying conditions.**

| Drying methods | Moisture (%) | Protein (%) | Fat (%) | Ash (%) | Fiber (%) | Carbohydrate (%) | $a_w$ |
|---|---|---|---|---|---|---|---|
| Fresh sample | 64.7±2.4b | 3.1±0.2a | 1.5±0.1a | 3.3±0.1a | 2.3±0.5a | 25.2±1.4a | 0.80±0.02b |
| HA | | | | | | | |
| 60˚C/210 min | 10.7±0.1a | 18.7±0.1b | 9.2±1.0b | 19.7±0.0b | 13.6±1.1b | 28.1±1.0b | 0.44±0.01a |
| 70˚C/120 min | 10.2±0.1a | 18.3±0.8b | 9.2±0.1b | 18.4±1.0b | 14.2±0.4b | 29.8±0.3b | 0.45±0.02a |
| 80˚C/100 min | 10.3±0.3a | 19.1±1.2b | 8.6±1.6b | 19.6±0.4b | 14.8±0.1b | 27.8±2.0b | 0.44±0.01a |
| MI | | | | | | | |
| 740 W/70 min | 10.5±0.1a | 18.6±0.1b | 9.9±0.7b | 18.8±1.1b | 14.2±0.0b | 27.9±0.7b | 0.49±0.30a |
| 780 W/45 min | 10.5±0.6a | 18.4±0.2b | 9.2±0.1b | 18.3±1.0b | 14.9±0.1b | 28.7±0.1b | 0.53±0.10a |
| 810 W/40 min | 10.1±0.4a | 19.2±1.0b | 9.7±0.5b | 19.7±0.1b | 14.1±0.0b | 27.2±0.2b | 0.47±0.00a |

*Values are given as mean ± SD from triplicate determinations.

**Different letters in the same column indicate significant differences ($p < 0.05$).

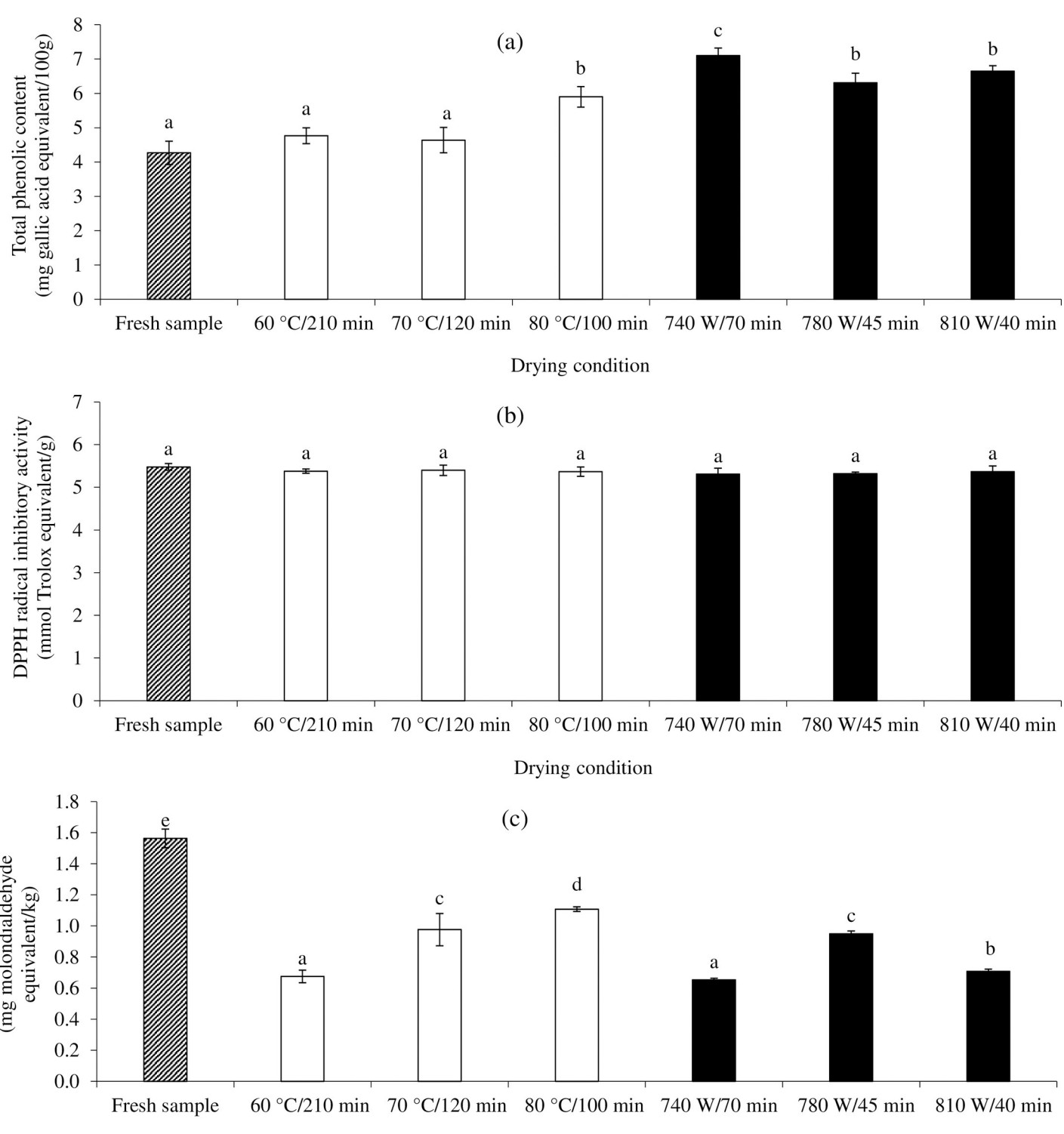

**Fig 3.** Total phenolic content (TPC) (a), DPPH radical scavenging activity (b), and TBARS (c) of *Tai-Pla* curry paste (fresh sample; pattern fill) and *Tai-Pla* curry powder (TCP) dried using hot air (HA) (□) and hybrid microwave-infrared drying (MI) (■) under different drying conditions. Different letters on the bars indicate significant differences (p<0.05).

phenolic compounds can be classified into two major forms, free and bound [41]. There were no significant difference in TPC among sample did by HA at 60°C and 70°C compared to the fresh sample (p>0.05). The thermal degradation of phenolic compounds may relate to this result. However, the greater TPC was observed in HA dried sample at 80°C (p<0.05). The availability of precursors of phenolic molecules by non-enzymatic inter-conversion between phenolic molecules may explain the formation of phenolic compounds at high temperature [42]. This was in agreement with the finding of Vega-Gálvez et al. [43] who reported a slight increase in TPC of red pepper after hot air drying at high temperature, in particular at 90°C. It can be seen that drying temperature has an important effect on phenolic remaining in dried product. For MI, drying at 740 W rendered the TCP with the highest TPC (p<0.05). The reduction of TPC at higher MI power intensity could be due to the heat-induced degradation. The opposite trend was reported by Hayat et al. [44] who found an increasing in TPC of citrus mandarin pomace with increasing microwave power (125–500 W/15 min). The different results may be due to the different raw material, microwave power, and drying time. Thus, MI power seemed to play an important role in the TPC retention during drying. Comparatively, sample dried with MI had higher TPC than that did with HA (p<0.05). This was due to a shorter drying time of MI, leading to a higher phenolic retained in dried sample. The result was in line with Özcan et al. [45] who found a higher TPC in microwave dried kiwi and pepino fruits than hot air counterpart. The presence of the TPC in the product may have helped to improve the oxidative stability during storage. The major phenolic compounds composed in the raw materials for production of TCP have been reported intensively, for instance, galangin in galangal [46], capsaicin in red chili pepper [47], allicin in garlic [48], isoorientin in lemongrass [49], piperine in black pepper [50], β-pinene and limonene in kaffir lime leaves [51], curcumin in turmeric [52] and quercetin in shallot [53].

**Free radical scavenging activity.** The DPPH radical scavenging activity of TCP made by HA and MI is depicted in Fig 3B. Dehydration temperatures and MI powers had no impact on radical scavenging activity (p>0.05). There were also non-significant difference among radical scavenging activity of all TCP and fresh sample (p>0.05). Although, antioxidative phenolic compounds varied depending on drying conditions (Fig 3A), similar radical scavenging activities of all sample were observed (Fig 3B). This behavior could be related to the degradation of certain original antioxidative compounds and simultaneous formation of new antioxidants (e.g. Maillard reaction products (see browning index in Table 2) which may decrease the

**Table 2. Color ($L^*$, $a^*$ and $b^*$ values) and browning index ($A_{294}$ and $A_{420}$) of *Tai-Pla* curry paste (fresh sample) and *Tai-Pla* curry powder dried using hot air (HA) and hybrid microwave-infrared drying (MI) under different drying conditions.**

| Drying methods | $L^*$ | $a^*$ | $b^*$ | $A_{294}$ | $A_{420}$ |
|---|---|---|---|---|---|
| Fresh sample | 27.65±0.05a | 8.52±0.24a | 28.96±0.83a | 45.24±0.01a | 10.03±0.00a |
| Hot-air | | | | | |
| 60°C/210 min | 43.82±0.08e | 11.17±0.10b | 43.31±0.17e | 79.98±0.02c | 18.96±0.00c |
| 70°C/120 min | 39.12±0.07b | 10.90±0.18b | 37.58±0.15b | 84.12±0.01d | 20.87±0.00d |
| 80°C/100 min | 39.97±0.20c | 11.17±0.20b | 38.59±0.34c | 86.18±0.02e | 20.83±0.00d |
| MI | | | | | |
| 740 W/70 min | 45.29±0.12f | 11.24±0.20b | 40.59±0.34c | 77.74±0.04b | 16.83±0.00b |
| 780 W/45 min | 43.26±0.12d | 11.04±0.04b | 39.65±0.48c | 76.02±0.00b | 16.71±0.00b |
| 810 W/40 min | 48.27±0.10g | 11.33±0.10b | 41.54±0.99c | 75.09±0.01b | 16.64±0.00b |

*Values are given as mean ± SD from triplicate determinations.

**Different letters in the same column indicate significant differences (p<0.05).

antioxidative capacity [54] or promote such activity [55]. A correlation between antioxidant activity and TPC has been reported during food dehydration [56]. Therefore, the net oxidative status of TCP was governed by the rate of degradation and formation of antioxidants during drying. However, numerous factors, such as drying method, type of extraction solvent, antioxidant assays, and interactions of several antioxidant reactions have been reported to affect the TPC and antioxidant activity of dried foods, which somehow resulted in a conflict data [42].

**TBARS.**    The effect of drying methods on the evolution of TBARS of TCP is shown in Fig 3C. TBARS increased with increasing HA temperature (p<0.05). This was attributed to the fact that unsaturated lipids are easily oxidized at higher temperature. Song et al. [57] reported that the drying temperature had progressively impact on lipid oxidation of lotus pollen. TBARS of pollen rapidly increased when drying at 50, 60, and 70°C in the first two hours of drying. For MI, TBARS value tended to increase with increasing MI power from 740 W to 780 W, then decreased afterward (Fig 3C). The changes in TBARS may be dependent on the level of heat generated by MI. The formation and accumulation of aldehydic lipid oxidation products could occur progressively with increasing MI power from 740 W to 780 W. However, at 810 W, a lower TBARS value was possibly due to the vaporization of volatile aldehydes and the interaction of reactive aldehydes with amines to form Maillard based products. De Pilli et al. [58] suggested that lipid oxidation of pasta increased with increasing microwave power during drying. From the results, the heat induced lipid oxidation and the newly formed antioxidants may involve in the net TBARS of TCP. Comparatively, the TBARS of MI dried powders were lower than HA dried samples (p<0.05). Although the same drying temperature between HA and MI were set, the different TBARS may result from the different drying time. The major changes in the powder structure during HA and MI may facilitate the accessible of oxygen to lipids, resulting in a greater lipid oxidation [59]. Interestingly, the fresh sample exhibited a greater TBARS than dried samples (p<0.05). This result might cause by the oxidation of lipid during curry paste preparation. The lipid oxidation of the fresh sample could be hastened during pasteurization at 90°C for 10 min.

**FTIR spectra.**    Similar FTIR spectra with different peak intensities were observed in all TCP (Fig 4), suggesting the identical compositions with different contents were found in all samples. From the results, the FTIR spectra can be grouped into two main regions. The first region, showing complex spectra, was located at 400–1,800 cm$^{-1}$ and the second region was located at 1,800–4,000 cm$^{-1}$. The complexity of the FTIR spectra especially in the bands at 400–1,800 cm$^{-1}$ was due to the presence of different ingredients used for production of TCP such as *Tai-Pla* sauce, spices, herbs, and fermented shrimp paste. Herein, the peak at 3,412 cm$^{-1}$ was typically found in all samples. Sanyal et al. [60] suggested that the peak region at 3,200–3,400 cm$^{-1}$ represented O-H stretching and this may be due to the combined effect of O-H groups of water and carbohydrates. The absorbance peak around 2,925 cm$^{-1}$ was related to typical vibration modes of the lipids fatty acids [61]. In addition, the absorbance band at 3,200–3,600 cm$^{-1}$ can be used to monitor the formation of hydroperoxide, a primary lipid oxidation product, [62] in which the highest bands were observed in TCP produced by HA at 70–80°C. This was somehow related to higher contents of TBARS, a secondary lipid oxidation product, of those TCP (Fig 3C). The peak region at 2,800–3,000 cm$^{-1}$ assigned to C-N stretching [60]. The peaks around 1000–1600 cm$^{-1}$ were also found in all samples, which were attributed to the presence of the amide of the characteristic protein bands [63]. The band at 1,033 cm$^{-1}$ region was assigned to C-N amines stretching [60]. Specifically, the detection of protein secondary structure is based on the amide I region composed of C = O stretching vibrations in the region of 1611–1690 cm$^{-1}$ [22]. Results indicated that proteins in TCP were in the denatured forms due to the effect of drying. The band at 1,410 cm$^{-1}$ region was assigned to O-H phenols bending [64], confirming the occurrence of the phenolic compounds in the TCP. Overall, the

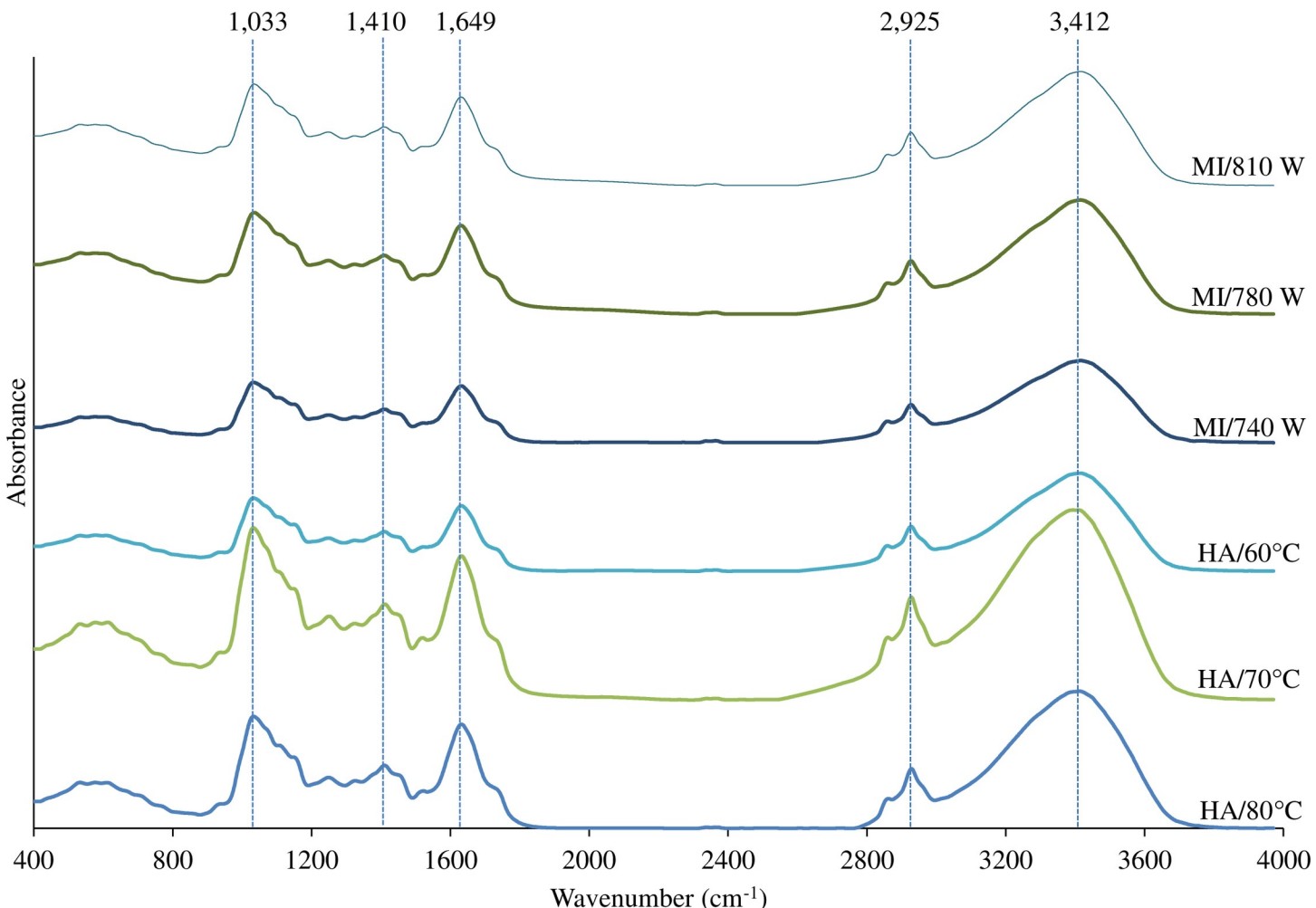

**Fig 4. FTIR spectra of *Tai-Pla* curry paste (fresh sample) and *Tai-Pla* curry powder (TCP) dried using hot air (HA) and hybrid microwave-infrared drying (MI) under different drying conditions.**

results indicated the presence of protein, lipid and its oxidation products, moisture, carbohydrate, and phenolic compounds in TCP.

## Physical characteristics

**Color and browning intensity.** Color is a crucial quality attribute of food, which influences the consumer's acceptance. The appearances of the fresh sample and final TCP prepared from the different drying conditions are shown in Fig 5.

Generally, the natural color of fresh *Tai-Pla* paste is roughly dark, caused by the dark color of *Tai-Pla* sauce, the key ingredient of this curry paste. Color parameters of TCP are displayed in Table 2. After drying, the dried powders had higher $L^*$, $a^*$, and $b^*$ values than that the original fresh sample (Table 2). The increased $L^*$ value was probably be due to the alteration of light reflection plane caused by water removal. A reduction of water content let the particles to freely flow and allowed more light to pass through the sample. The increased $a^*$ and $b^*$ values were mainly due to the evolution of browning reactions, e.g. Maillard reaction, caramelization, and polyphenol oxidation. Sharifian et al. [65] reported that the changes in color of dried food

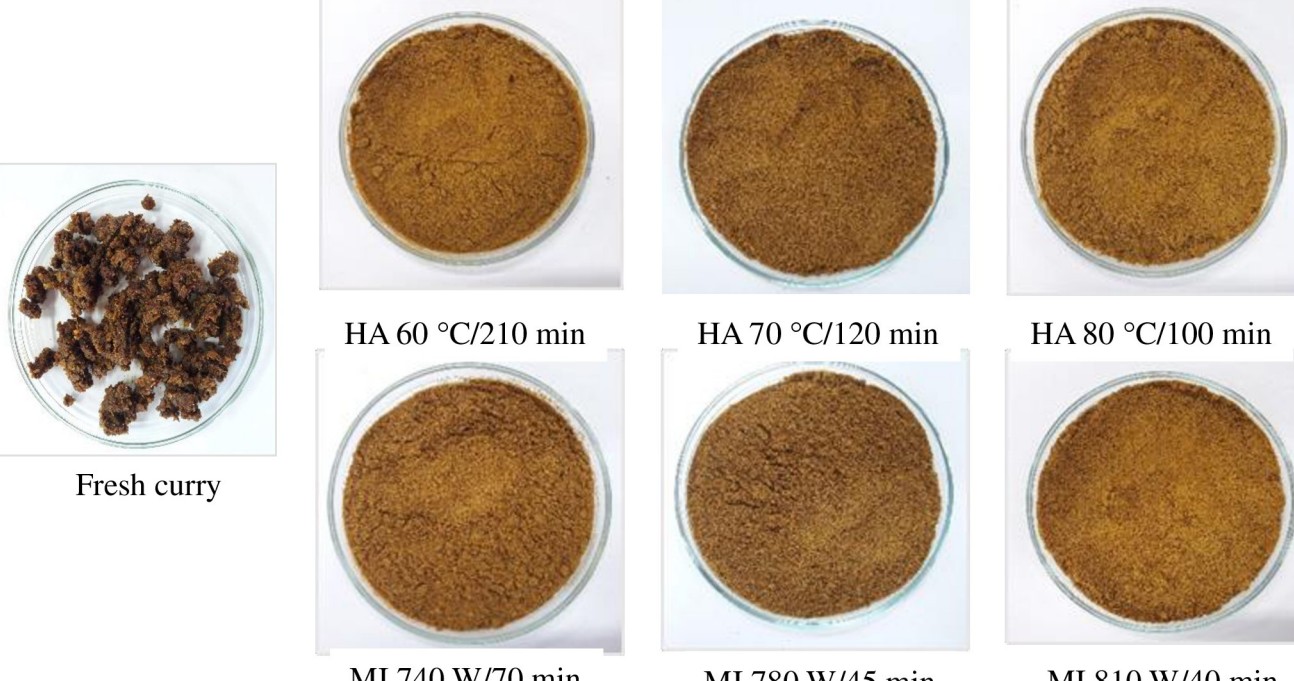

**Fig 5. Appearances of *Tai-Pla* curry paste (fresh sample) and *Tai-Pla* curry powder (TCP) dried using hot air (HA) and hybrid microwave-infrared drying (MI) under different drying conditions.**

might contribute to the complex chemical reactions such as pigment oxidation, chemical degradation, and browning reaction.

For HA, a lower drying temperature (60°C) rendered the sample with greater $L^*$ and $b^*$ values (p<0.05). With increasing temperature in HA, $L^*$ and $b^*$ values decreased but $a^*$ value remained constant (Table 2). No significant differences in $a^*$ values were detected among HA dried samples (p>0.05). For MI, no significant differences in $a^*$ and $b^*$ values were observed among samples (p>0.05) whereas the $L^*$ values were slightly different among samples. This finding was in accordance with Inchuen et al. [66] for microwave-dried red curry, Pereira et al.

**Table 3. Bulk density and wettability of *Tai-Pla* curry paste (fresh sample) and *Tai-Pla* curry powder dried using hot air (HA) and hybrid microwave-infrared drying (MI) under different drying conditions.**

| Drying methods | Bulk density (g/mL) | Wettability (%) |
|---|---|---|
| Fresh sample | 1.26±0.01d | 8.95±0.72a |
| HA | | |
| 60°C/210 min | 0.54±0.01ab | 24.39±0.35b |
| 70°C/120 min | 0.51±0.02a | 24.02±0.55b |
| 80°C/100 min | 0.57±0.02b | 24.10±0.80b |
| MI | | |
| 740 W/70 min | 0.61±0.02c | 25.39±0.15c |
| 780 W/45 min | 0.57±0.12bc | 26.70±0.06c |
| 810 W/40 min | 0.62±0.02c | 24.87±0.14b |

*Values are given as mean ± SD from triplicate determinations.

**Different letters in the same column indicate significant differences (p<0.05).

[67] for combined microwave/hot-air-dried banana, Soysal [68] for microwave-dried parsley, and Maskan [69] for microwave-dried banana.

Browning effects are mainly contributed to the quality deterioration of dried food products caused by thermally induced browning reactions [70]. The absorption peak at a wavelength of 294 nm ($A_{294}$) indicates the presence of intermediate compounds whereas the intensity at 420 nm ($A_{420}$) indicates the formation of brown pigments of Maillard reaction or caramelization. The browning index of TCP are presented in Table 2. The TCP showed the greater $A_{294}$ and $A_{420}$ than the fresh paste, demonstrating the evolution of browning reaction during drying. An increase in HA temperature caused a significant increase in the formation of Maillard intermediate products ($p < 0.05$). Non-enzymatic browning reactions particularly Maillard reaction and caramelization are temperature dependent. The higher the temperature, the greater the browning intensity [65]. Vega-Gálvez et al. [43] reported that the amounts of reducing sugars and amino acids in the raw materials played an important role in the degree of Maillard reaction during the drying process. On the flip side, the MI power had no impact on the accumulation of both intermediate products and brown pigment ($p > 0.05$). Comparatively, the browning intermediates and brown pigments in traditional HA dried samples were greater than those in MI dried samples, suggesting a better color protective effect of the MI. Since the similar drying temperatures were employed between HA and MI, heat generated during MW by molecular friction caused the greater heat load inside the sample. In addition, MI quickly produced heat and effectively removed water at surface. This can shorten the drying time and hence reduce the degree of browning reactions. TCP produced by HA had a higher browning index (Table 2) whereas TCP produced by MI tended to have a higher TPC (Fig 3A). Both browning intermediate/final products and phenolic compounds can function as antioxidant. So, the net free radical scavenging activity of TCP was similar among MI and HA (Fig 3B).

**Bulk density.** The bulk density of food powder can typically present its textural characteristics. The bulk density of fresh sample was 1.26 g/mL (Table 3). A lower bulk density was observed in all TCP, ranging between 0.51 g/mL to 0.61 g/mL (Table 3). The larger particle sizes with higher void spaces could originally lower the bulk density of dried sample [71]. Among HA dried powders, TCP dried at 80˚C showed the highest bulk density, followed by samples dried at 60˚C and 70˚C, respectively (Table 3). No significant difference in bulk density was noticed among MI dried samples ($p > 0.05$), suggesting a comparable textural characteristic. This was in accordance with the finding of Koç and Çabuk [72] who found a constant bulk density of egg white powder dried with different microwave powers (120–350 W). TCP dried with MI method showed higher bulk density than those HA dried samples ($p < 0.05$). A higher bulk density may contribute to a decrease in the inter-particle void volume of dried powder with larger contact surface area per unit volume. Changes in bulk density and porosity of powder were influenced by powder characteristics such as particle size [73].

**Wettability.** Wettability presents the reliable criterion of powder behavior in aqueous solution. The sinkability, dispersability, and wettability of powder are underwent after the dissolution steps [74]. The wettability of the TCP made by different drying methods is shown in Table 3. Generally, all TCP had greater water wettability than the fresh sample ($p < 0.05$). This result could contribute to the higher degree of macromolecular disorganization of the dried particles as affected by drying process. There were no significant difference in the wettability among HA dried TCP ($p > 0.05$) (Table 3). The MI dried TCP had higher wettability than HA dried TCP ($p < 0.05$). This may be due to the lower structure disruption of MI dried sample caused by the shorter drying time. It should be noted that drying using MI output power of 810 W resulted in a slight decrease in the wettability of TCP. Koç and Çabuk [72] reported that the wettability of egg white powder increased with increasing microwave power from 120 W

to 460 W and subsequently decreased with further increasing microwave power from 460 W to 600 W. From the results, the wettability of TCP was governed by drying condition.

## Conclusion

The effects of two different drying methods (HA and MI) on quality of TCP were evaluated. Results suggested that the color, bulk density, wettability, TPC, radical scavenging activity, and lipid oxidation of final TCP were dependent on drying conditions. The optimal MI condition for TCP preparation was observed at 810 W for 40 min, which was faster than the traditional HA for 5 times. At this condition, TCP had the superior physico-chemical characteristics (TPC, oxidative stability, color, and bulk density) to the traditional HA without a negative effect on the proximate compositions. Therefore, MI is a promising method for TCP production after considering the overall quality characteristics. Further studies are recommended to determine the sensory quality, flavor profile, and storage stability of the TCP.

## Acknowledgments

Food Technology and Innovation Research Center of Excellence, Walailak University was acknowledged for providing the scientific and technological equipment for this research.

## Author Contributions

**Conceptualization:** Worawan Panpipat, Mudtorlep Nisoa, Ling-Zhi Cheong, Manat Chaijan.

**Data curation:** Warongporn Choopan.

**Formal analysis:** Worawan Panpipat, Ling-Zhi Cheong.

**Funding acquisition:** Worawan Panpipat.

**Methodology:** Worawan Panpipat, Manat Chaijan.

**Resources:** Worawan Panpipat, Mudtorlep Nisoa.

**Supervision:** Worawan Panpipat, Manat Chaijan.

**Writing – original draft:** Warongporn Choopan, Worawan Panpipat, Mudtorlep Nisoa, Ling-Zhi Cheong, Manat Chaijan.

**Writing – review & editing:** Warongporn Choopan, Worawan Panpipat, Mudtorlep Nisoa, Ling-Zhi Cheong, Manat Chaijan.

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
