## [Decision Letter · Decision Letter 0]

10 May 2021

PONE-D-21-10233

Physico-chemical aspects of Thai fermented fish viscera, Tai-Pla, curry powder processed by hot air drying and hybrid microwave-infrared drying

PLOS ONE

Dear Dr. Panpipat,

Thank you for submitting your manuscript to PLOS ONE. After careful consideration, we feel that it has merit but does not fully meet PLOS ONE’s publication criteria as it currently stands. Therefore, we invite you to submit a revised version of the manuscript that addresses the points raised during the review process.

We look forward to receiving your revised manuscript.

Kind regards,

C. Anandharamakrishnan

Academic Editor

PLOS ONE

Journal Requirements:

2. We understand that you purchased ingredients from local markets for this study. In your Methods section, please provide additional regarding the source of this material. Please provide the geographic coordinates and names of the purchase locations (e.g., stores, markets), if available, as well as any further details about the purchased items (e.g., lot number, source origin, description of appearance) to ensure reproducibility of the analyses.

"This research was financially supported by Research and Researchers for Industries (RRI) program and Shaw Processing Food Co. Ltd. [Grant No. MSD61I0053] and the new strategic research project (P2P), Walailak University, Thailand. The funders had no role in study design, data collection and analysis, decision to publish, or preparation of the manuscript."

We note that you received funding from a commercial source: Shaw Processing Food Co. Ltd.

Reviewers' comments:

Reviewer's Responses to Questions

**Comments to the Author**

1. Is the manuscript technically sound, and do the data support the conclusions?

Reviewer #1: Partly

Reviewer #2: Yes

Reviewer #3: Partly

Reviewer #4: Yes

Reviewer #5: No

Reviewer #6: Yes

Reviewer #7: Partly

Reviewer #8: Partly

2. Has the statistical analysis been performed appropriately and rigorously? 

Reviewer #1: Yes

Reviewer #2: Yes

Reviewer #3: Yes

Reviewer #4: Yes

Reviewer #5: No

Reviewer #6: Yes

Reviewer #7: No

Reviewer #8: No

3. Have the authors made all data underlying the findings in their manuscript fully available?

Reviewer #1: Yes

Reviewer #2: Yes

Reviewer #3: Yes

Reviewer #4: Yes

Reviewer #5: No

Reviewer #6: Yes

Reviewer #7: No

Reviewer #8: Yes

4. Is the manuscript presented in an intelligible fashion and written in standard English?

Reviewer #1: Yes

Reviewer #2: Yes

Reviewer #3: No

Reviewer #4: Yes

Reviewer #5: No

Reviewer #6: Yes

Reviewer #7: Yes

Reviewer #8: Yes

5. Review Comments to the Author

Reviewer #1: The manuscript entitled Physico-chemical aspects of Thai fermented fish viscera, Tai-Pla, curry powder processed by hot air drying and hybrid microwave-infrared drying finds to be interesting. Further some major revision is required before final acceptance.

General Comments:

1. In this work, author used hybrid MI drying method for drying of Thai fermented fish viscera, Tai-Pla, curry powder (TCP). But throughout the manuscript only microwave power level varies and nothing is mentioned about Infra-red drying. In Fig 1 also there is no provision of Infra-red heating, then how this method become hybrid drying.

2. There is no comparison of energy consumption of HA and MI in the manuscript.

3. The applied Drying time in MI method is higher range (above 40 min). Normally drying in microwave upto 10 min maximum.

Specific Comments:

1. In Introduction give more recent references related with MI drying method.

2. Line no. 120-121, On what basis final moisture content 12% and water activity 0.6 was chosen. Whether its safe moisture content of storage of TCP. If yes, then give reference for it.

3. What is the sample size dry in both the method?

4. In this research work, TCP dried at three different temperatures 60, 70 and 800C. On what basis this range was selected. Please mentioned in the manuscript.

5. Line 152-153; please specify why TBRS is important to study in this research work.

6. Line 162-163, why author want to use FTIR in this study. Please mention? On what basis the FTIR range was selected?

7. Line 170-171, How Browning Intensity influence the product quality. Please specify

Reviewer #2: The paper ”Physico-chemical aspects of Thai fermented fish viscera, Tai-Pla, curry powder processed by hot air drying and hybrid microwave-infrared drying” aimed to comparatively investigate the effect of hot air drying (HA) and hybrid microwave-infrared drying (MI) on physico-chemical characteristics of Thai fermented fish viscera, Tai-Pla, curry powder (TCP). The subject is of great interest for the industrials willing to reinterpret the traditional technology into convenient innovative alternatives.

There are few aspects that need attention from the authors.

Indicate how many replicates were taken in this study.

When a product is designed for further industrial scale production, the readers are willing to see the sensory analysis results. Did you perform sensory data or are you able to perform it and include the results in this manuscript?

For better point the advantage of MI treatment, authors should perform also some stability tests and specify the shelf life of the obtained powders. Is it possible to perform also the microbiological analysis? At least the common ones ? You can also insert a discussion related to the influence of fat content to product stability.

Please argument the necessity of antioxidant activity (AA) and correlate the your AA results to total phenolic content, not just simply say they correlate based on literature analysis, because it is not always true. So please check this with your results. Why did authors decide not to include also the individual phenolic compound analysis?

The discussion on FTIR results is poor. Please improve this part otherwise is not useful for the readers.

Reviewer #3: The manuscript deals with physico-chemical aspects of Thai fermented fish viscera, Tai-Pla, curry powder processed by hot air drying and hybrid microwave-infrared drying.

The English language must be revised.

Please separate values from units, e.g. “60 ºC” not “60ºC”.

Please number all sections.

Abstract

This section is vague. Please present your main results.

Introduction

The topics must be better linked.

Materials and methods

Line 107- “Thereafter, all ingredients with the specified proportion were mixed and coarsely ground for 10 min using a grinder (MK 5087M Panasonic Food Processor, Selangor Darul Ehsan, Malaysia) to obtain fresh curry paste.”??amounts used of each ingredient??

Line 109- “The fresh curry paste was then pasteurized at 90ºC for 10 min.”??how was the sample pasteurized?amount used??

Line 114- “Two drying techniques including HA and MI were used to prepare Tai-Pla curry powder. The fresh samples were uniformly spread in the tray with the thickness of 0.5 cm and subjected to dry using a traditional HA drier (tray drier) or an MI drier. The MI drier used in this study was developed by the Center of Excellence in Plasma Science and Electromagnetic Waves, Walailak University (Fig. 1). The MI output powers were adjusted to 740, 780, and 810 W in order to meet the drying temperature of 60ºC, 70ºC, and 80ºC as done by a traditional HA drier.”??used conditions??tray dryer air speed??

Line 115- “The fresh samples were uniformly spread in the tray with the thickness of 0.5 cm and subjected to dry using a traditional HA drier (tray drier) or an MI drier.”??amount used??

Line 123- “The obtained TCP was packed in an aluminum foil laminated bag to prevent moisture adsorption and kept in an auto desiccator at room temperature for 24 h.”??packaging dimensions??

Line 167- “Colorimetric values of the samples were measured using a Hunterlab colorimeter (Hunter Assoc. Laboratory; VA, USA). The L*, a*, and b* values were recorded.”???illuminant used??ºobserver??calibration??

Line 184- “Wettability”??or solubility in water??

Results and discussion

This section has lack of depth and must be improved.

Line 251- “From the results, TCP contributed not only for taste and flavor enhancers, but also played a part in an extra-source of nutrients.”?? Aroma???flavor??measured??

Particle size??

Figure 1- Please define each component.

Figure 4- Please add wavenumber in each peak.

Conclusion

Line 451- “Results suggested that TCP prepared by MI method showed the superior physico-chemical characteristics to the traditional HA.”???superior??in which results??

References

37 references have more than 5 years. Please update your list of references.

Reviewer #4: In the present study titled “Physico-chemical aspects of Thai fermented fish viscera, Tai-Pla, curry powder processed by hot air drying and hybrid microwave-infrared drying” is presented in detailed a well elaborated research which evaluated the properties of Thai fermented fish viscera, Tai-Pla, curry powder (TCP) dryed on different approaches: hot air drying (HA) and hybrid microwave-infrared drying (MI). The properties of both fresh and dried powders were evaluated by proximate composition, aw, total phenolic content, DPPH radical scavenging activity, TBARS, FT-IR spectroscopy and physical characteristics (colour, browning intensity, bulk density, wetability).

The research led to the identification of the best drying procedure of the fresh pasta with the desired properties, namely the hybrid microwave-infrared drying (MI).

In general the data are strong, and convincingly shows that the hybrid microwave-infrared drying (MI) approach could be used as a drying procedure to obtain good quality food products. The manuscript is well written, concise and the appropriate analyses are performed.

Overall, this is a well performed study that I consider that is important and represent a new strategy to conveniently obtain Thai fermented fish viscera, Tai-Pla, curry powder (TCP) with good quality characteristics.

The authors need to address the below comments to strengthen the quality of the manuscript:

1. Please insert the characterization methods used in the present study in the Abstract (e.g. FT-IR, DPPH assay).

2. Please replace the phrases from line 37 and 455: „Therefore, MI was a promising drying

3. technique to reduce the drying time and improve the overall quality of TCP.” By „Therefore, MI is a promising drying technique to reduce the drying time and improve the overall quality of TCP.”

4. In the preparation method of the pasta (in Materials and methods) please include the mass percentage of the main ingredients used to obtain the product.

Reviewer #5: In this manuscript, the authors compared the effect of hot air drying (HA) and hybrid microwave-infrared drying (MI) on physico-chemical characteristics of Thai fermented fish viscera, Tai-Pla, curry powder (TCP). The data in this article is not solid and well analyzed. Besides, the article doesn't present the application superiority of MI thoroughly, in other words doesn't fit with the average originality found of PLOS ONE. I recommend to not publish the article.

Major comments

The data succeed to show the different effect of HA and MI, but the experiments are too superficial, as well the test of antioxidant activities. According to the introduction, the application of TCP mainly gives the Thai food a special flavor and aroma, which should be concerned in this article because the HA and MI processing would change the flavor of TCP. In the overall the text is well written but is superficial and is out of the journal standards.

Detailed comments

1. The authors developed a hybrid microwave-infrared drier but the schematic diagram (Fig. 1) didn’t show how the hybrid microwave-infrared drier works.

2. SEM is recommended to show the influence of these physical processes on tissues of TCP during drying.

3. Line 36: better change “between … to …” to “from … to …”.

4. Line 61: “a” should be “its” or deleted.

5. Line 76-77: Please check this sentence and revise.

6. Line 163: FTIR spectra of fresh samples can’t be found in Figure 4.

7. Line 251-252: Please check this sentence and revise.

8. Line 271: Please check this sentence and revise.

9. Line 380-381: Please check this sentence and revise.

10. Line 411-412: Please check this sentence and revise.

11. Line 421: “P<0.05” is wrong.

12. Please replace “governed” with another word in line 30, line 307, and line 441.

13. Line 51: “cuisin” should be “cuisine”.

14. Please add a comma before the second subject in line 215-217, line 217-218, and line 402-403.

15. Please mark “a, b, c” in the Figure 3 according to “Fig. 3a, Fig. 3b, Fig. 3c” in line 302-313.

Reviewer #6: In this manuscript, Choopan et al. compared the hot air (HA) and hybrid microwave-infrared (MI) drying ways on the Physico-chemical characteristics of TCP. They have demonstrated that MI drying with a condition of 810 W for 40 min effectively reduced the drying time by five-fold. They also showed that MI-dried TCP had the lowest browning index, the highest lightness, higher phenolic content, and lower TBARS, indicating improved overall quality. The manuscript has convincing data to support their conclusion.

The concern I have is whether MI drying affects the flavor of TCP. Is it possible to evaluate the flavor?

Reviewer #7: The paper presents an application of Physico-chemical aspects for Thai fermented fish viscera, Tai-Pla, curry powder processed. It is a topic of interest to the researchers in the related areas but this paper needs improvement before acceptance for publication. My detailed comments are as follows:

1. The sample source and size are both important for this paper, and please provide more details about the sample information in the section Tai-Pla curry paste preparation.

2. Instrument model and manufacturer used for the drying process are not provided in the section Drying experiment. Please provide more details about these information.

3. In the sections “Total phenolic content (TPC) and DPPH radical scavenging activity”, “Thiobarbituric acid reactive substances (TBARS)”, and “Fourier transform infrared (FTIR) spectroscopy”, only 20 g (i.e., 10g for Total phenolic content (TPC) and DPPH radical scavenging activity and 10 g for Thiobarbituric acid reactive substances (TBARS)) is not enough for us to consider the robustness of your chemical measurements. You must provide more samples to ensure it.

4. Please provide more details in the section Fourier transform infrared (FTIR) spectroscopy and Color.

5. In the section “Statistical analysis”, please provide the sample size and the sample category for the data analysis.

6. The increase of TPC from the fresh sample to the drying sample is not reasonable in the Figure 3. Please show us more reason or information about it .

Reviewer #8: The manuscript entitled “Physico-chemical aspects of Thai fermented fish viscera, Tai-Pla, curry powder processed by hot air drying and hybrid microwave-infrared drying” investigated the effect of hot air drying (HA) and hybrid microwave-infrared drying (MI) on physico-chemical characteristics of Thai fermented fish viscera, Tai-Pla, curry powder (TCP). The present manuscript requires major revision before considering for the acceptance.

1. Add references in the line 54-64

2. Mention the quantity of each ingredients taken for the preparation of Tai-Pla curry paste (Line 102-104)

3. The authors mentioned that the fresh curry paste was pasteurized at 90oC for 10 min. There is any standard pre-optimized protocol/references available?

4. In case of MI how output power level calculation was done? And what basis the power level of 740, 780, and 810 W was chosen?

5. Elaborate the standard methodology (AOAC) followed to determine moisture, protein, fat, ash, and carbohydrate

6. In line 168, add the description for color value L*, a* and b*

7. In line 207 author have mentioned drying time 70oC and 80 oC had the similar drying time of 120 min. Why so? If there is difference in 10 oC could achieved at 120 min?

8. What would be the final temperature achieved at 740W/70 min, 780W/45 min and 810W/40 min?

9. Rewrite the whole section in results part -Free radical scavenging activity, TBARS- precisely with the obtained values and compare with other studies

10. Explain the effect of drying on functional group in FTIR-Rewrite the paragraph

11. Revise the conclusion part as per the obtained results

12. Need to add statistical design or experimental design

13. Fig 1. label the each parts

14. Fig 4 & 5 labelling is inappropriate

6. PLOS authors have the option to publish the peer review history of their article (what does this mean?). If published, this will include your full peer review and any attached files.

Reviewer #1: No

Reviewer #2: No

Reviewer #3: No

Reviewer #4: No

Reviewer #5: No

Reviewer #6: No

Reviewer #7: No

Reviewer #8: **Yes: **VENKATACHALAPATHY N

---

## [Author Response · Author response to Decision Letter 0]

21 May 2021

Response to Reviewers

All points raised by the reviewers were carefully addressed and answered point-by-point. A revision was made in highlighted red fonts. The revised manuscript was carefully prepared to meet PLOS ONE's style requirements.

Journal Requirements:

Ans: The revised manuscript was carefully prepared to meet PLOS ONE's style requirements.

2. We understand that you purchased ingredients from local markets for this study. In your Methods section, please provide additional regarding the source of this material. Please provide the geographic coordinates and names of the purchase locations (e.g., stores, markets), if available, as well as any further details about the purchased items (e.g., lot number, source origin, description of appearance) to ensure reproducibility of the analyses.

Ans: The geographic coordinate and name of the purchase location were given. The details about the purchased items were also given.

"This research was financially supported by Research and Researchers for Industries (RRI) program and Shaw Processing Food Co. Ltd. [Grant No. MSD61I0053] and the new strategic research project (P2P), Walailak University, Thailand. The funders had no role in study design, data collection and analysis, decision to publish, or preparation of the manuscript."

We note that you received funding from a commercial source: Shaw Processing Food Co. Ltd.

Ans: We got the co-funding grant from Research and Researchers for Industries (RRI) program and Shaw Processing Food Co. Ltd. [Grant No. MSD61I0053]. The main funding came from the RRI program and a partial funding from a commercial source: Shaw Processing Food Co. Ltd. This grant aims to encourage the graduate student to work with industry and the results can be published as signed in the grant contract. Therefore, we confirmed that this does not alter our adherence to all PLOS ONE policies on sharing data and materials. The Competing Interests Statement "This does not alter our adherence to PLOS ONE policies on sharing data and materials.” was added in the cover letter.

Reviewer #1: The manuscript entitled Physico-chemical aspects of Thai fermented fish viscera, Tai-Pla, curry powder processed by hot air drying and hybrid microwave-infrared drying finds to be interesting. Further some major revision is required before final acceptance.

General Comments:

1. In this work, author used hybrid MI drying method for drying of Thai fermented fish viscera, Tai-Pla, curry powder (TCP). But throughout the manuscript only microwave power level varies and nothing is mentioned about Infra-red drying. In Fig 1 also there is no provision of Infra-red heating, then how this method become hybrid drying.

Ans: In this study, the infrared heater is fixed at 500 watts. The Fig. 1 was revised to indicate the presence of infrared heater in the oven. So, the use of an adjustable power microwave heating (740 W, 780 W, and 810 W) coupled with a fixed power infrared heating (500 W) can be recognized as a hybrid drying. The information regarding this issue was added in the Abstract and Materials and methods. 

2. There is no comparison of energy consumption of HA and MI in the manuscript.

Ans: The specific energy consumption of HA and MI was given in Fig. 2c. The Method and the Discussion regarding the energy consumption were also given.

3. The applied Drying time in MI method is higher range (above 40 min). Normally drying in microwave upto 10 min maximum.

Ans: In this study, the power of microwave was set at 740-810 W and the target moisture content was around 12%. In comparison with previous report using the same MI machine, the drying time for stink bean seed (600 W/target moisture content of 15%) was about 6-10 h (Nisoa M, Wattanasit K, Tamman A, Sirisathitkul Y, Sirisathitkul C. Microwave drying for production of rehydrated foods: A case study of stink bean (Parkia speciosa) seed. Appl Sci. 2021;11:2918.). So, the drying time using MI was dependent on the applied power, target moisture content, and the type and nature of raw material.

Specific Comments:

1. In Introduction give more recent references related with MI drying method.

Ans: The recent references related with MI drying method were added.

2. Line no. 120-121, On what basis final moisture content 12% and water activity 0.6 was chosen. Whether its safe moisture content of storage of TCP. If yes, then give reference for it.

Ans: In Thailand, the water activity (aw) of <0.65 and the moisture content of <13% (w/w) are the standards set for seasoning powder [15]. To comply with this standard and to ensure the food safety, the drying was proceeded until the final moisture content was reached �12% with the aw of �0.6. This was given in the “Drying experiment”.

3. What is the sample size dry in both the method?

Ans: The sample size was given for both methods. “The fresh samples (1000 g) were uniformly spread in the tray with the thickness of 0.5 cm and subjected to dry using a traditional HA drier (tray drier) or an MI drier.”

4. In this research work, TCP dried at three different temperatures 60, 70 and 800C. On what basis this range was selected. Please mentioned in the manuscript.

Ans: From the report of our colleges using the same HA (tray drier), they suggested the maximum temperature for drying whole stink bean at 70�C. They stated that if the temperature is higher than 70�C, the dehydration rate may be too high for that product. So, in this study we tried to vary the drying temperatures using the basis of 70�10�C (60, 70, and 80�C) for TCP. This was mentioned in the manuscript “It has been reported that the maximum temperature used for drying the whole stink bean (Parkia speciosa) seed was at 70�C using the same HA (tray drier) [5]. So, in this study, the drying temperatures were varied on the basis of 70�10 �C which were 60 �C, 70 �C, and 80 �C for TCP.”

5. Line 152-153; please specify why TBRS is important to study in this research work.

Ans: We stated in the Materials and method that “TBARS analysis is the most widely used method to determine the secondary lipid oxidation products [20]. Due to the presence of some lipid content in the Tai-Pla curry paste, the lipid oxidation can be taken placed during drying. Herein, the lipid oxidation was monitored using the TBARS.”

6. Line 162-163, why author want to use FTIR in this study. Please mention? On what basis the FTIR range was selected?

Ans: The detail about the FTIR was given in the Materials and methods. “The FTIR spectroscopy is a vibrational spectroscopic technique that can be used to characterize the substances by identifying their functional groups presented [22]. FTIR spectra (400-4000 cm-1 with the resolution of 4 cm-1 at the average of 16 scans) of the TCP were obtained using a horizontal Attenuated Total Reflectance (ATR) Trough plate crystal cell (45° ZnSe; 80 mm long, 10 mm wide and 4 mm thick) (Pike Technology, Inc., Madison, WI, USA) equipped with a Bruker Model Vector 33 FTIR spectrometer (Bruker Co., Ettlingen, Germany) at room temperature. Analysis of spectral data was carried out using the OPUS 3.0 data collection software program.”

7. Line 170-171, How Browning Intensity influence the product quality. Please specify

Ans: The effect of browning intensity on the product quality was given and this section was revised accordingly. “The browning intensity can be used to monitor the formations of the intermediate and final products of the Maillard reaction in the TCP. The presence of the Maillard reaction products (MRPs) affected both color and antioxidant activity of the food products [23]. The UV absorbance at 294 nm (A294) was often used to indicate the intermediate MRPs while the final MRPs was monitored by the absorbance at 420 nm (A420)…..” 

Reviewer #2: The paper ”Physico-chemical aspects of Thai fermented fish viscera, Tai-Pla, curry powder processed by hot air drying and hybrid microwave-infrared drying” aimed to comparatively investigate the effect of hot air drying (HA) and hybrid microwave-infrared drying (MI) on physico-chemical characteristics of Thai fermented fish viscera, Tai-Pla, curry powder (TCP). The subject is of great interest for the industrials willing to reinterpret the traditional technology into convenient innovative alternatives.

There are few aspects that need attention from the authors.

Indicate how many replicates were taken in this study.

Ans: Three replications were taken in this study as stated in the Statistical analysis section. “A completely randomized design was used in this study. The data were expressed as means ± standard deviations (SD) of three replications (n =3) for all analyses.”

When a product is designed for further industrial scale production, the readers are willing to see the sensory analysis results. Did you perform sensory data or are you able to perform it and include the results in this manuscript?

Ans: Thank you for your invaluable suggestion. This study aimed to explore the physico-chemical aspects of the TCP. We will do the sensory analysis in the future. However, from our own observation, the sensory aspects (odor, flavor, taste) of the TCP were remained unchanged. The recommendation regarding this issue was added in the Conclusion. “Further studies are recommended to determine the sensory quality, flavor profile, and storage stability of the TCP.”

For better point the advantage of MI treatment, authors should perform also some stability tests and specify the shelf life of the obtained powders. Is it possible to perform also the microbiological analysis? At least the common ones ?

Ans: Thank you for your invaluable suggestion. We will do the stability test including the microbiological analysis in the future. However, the recommendation regarding this issue was added in the Conclusion. “Further studies are recommended to determine the sensory quality, flavor profile, and storage stability of the TCP.”

 You can also insert a discussion related to the influence of fat content to product stability.

Ans: The discussion regarding the fat content to product stability, especially oxidative stability, was included. 

“TBARS analysis is the most widely used method to determine the secondary lipid oxidation products [20]. Due to the presence of some lipid content in the Tai-Pla curry paste, the lipid oxidation can be taken placed during drying. Herein, the lipid oxidation was monitored using the TBARS.”

“The presence of fat in the product may cause the oxidative instability during storage.”

“The effect of drying methods on the evolution of TBARS of TCP is shown in Fig. 3c. TBARS increased with increasing HA temperature (p<0.05). This was attributed to the fact that unsaturated lipids are easily oxidized at higher temperature.”

Please argument the necessity of antioxidant activity (AA) and correlate the your AA results to total phenolic content, not just simply say they correlate based on literature analysis, because it is not always true. So please check this with your results. 

Ans: A revision was made. “Although, antioxidative phenolic compounds varied depending on drying conditions (Fig. 3a), similar radical scavenging activities of all sample were observed (Fig. 3b). This behavior could be related to the degradation of certain original antioxidative compounds and simultaneous formation of new antioxidants (e.g. Maillard reaction products (see browning index in Table 2) which may decrease the antioxidative capacity [54] or promote such activity [55]. A correlation between antioxidant activity and TPC has been reported during food dehydration [56]. Therefore, the net oxidative status of TCP was governed by the rate of degradation and formation of antioxidants during drying. However, numerous factors, such as drying method, type of extraction solvent, antioxidant assays, and interactions of several antioxidant reactions have been reported to affect the TPC and antioxidant activity of dried foods, which somehow resulted in a conflict data [42].”

Why did authors decide not to include also the individual phenolic compound analysis?

Ans: Thank you very much for you invaluable suggestion. We will try to detect the phenolic profile of our product in the future. At present, we have no equipment to do it. However, major phenolic compounds found in the raw materials for production of TCP were added using the previous reports from the literatures. “The presence of the TPC in the product may have helped to improve the oxidative stability during storage. The major phenolic compounds composed in the raw materials for production of TCP have been reported intensively, for instance, galangin in galangal [46], capsaicin in red chili pepper [47], allicin in garlic [48], isoorientin in lemongrass [49], piperine in black pepper [50], β-pinene and limonene in kaffir lime leaves [51], curcumin in turmeric [52] and quercetin in shallot [53].”

The discussion on FTIR results is poor. Please improve this part otherwise is not useful for the readers.

Ans: The discussion on FTIR was intensively revised (See FTIR spectra). 

Reviewer #3: The manuscript deals with physico-chemical aspects of Thai fermented fish viscera, Tai-Pla, curry powder processed by hot air drying and hybrid microwave-infrared drying.

The English language must be revised.

Ans: English language was rechecked and polished.

Please separate values from units, e.g. “60 ºC” not “60ºC”.

Ans: Done throughout the text.

Please number all sections.

Ans: We prepared the manuscript following the PLOS ONE style templates.

Abstract

This section is vague. Please present your main results.

Ans: Abstract was revised intensively.

Introduction

The topics must be better linked.

Ans: The Introduction was revised and all topics in the Introduction were linked.

Materials and methods

Line 107- “Thereafter, all ingredients with the specified proportion were mixed and coarsely ground for 10 min using a grinder (MK 5087M Panasonic Food Processor, Selangor Darul Ehsan, Malaysia) to obtain fresh curry paste.”??amounts used of each ingredient??

Ans: The mass percentage of the main ingredients used for production of Tai-Pla curry was given.

Line 109- “The fresh curry paste was then pasteurized at 90ºC for 10 min.”??how was the sample pasteurized?amount used??

Ans: The information was added. “The fresh curry paste (1,000 g) was then pasteurized at 90 �C for 10 min [14] in a controlled temperature Hanabishi HGP160S electric pan (Hanabishi Electric Co., Ltd., Bangkok Thailand) under continuous stirring.”

Line 114- “Two drying techniques including HA and MI were used to prepare Tai-Pla curry powder. The fresh samples were uniformly spread in the tray with the thickness of 0.5 cm and subjected to dry using a traditional HA drier (tray drier) or an MI drier. The MI drier used in this study was developed by the Center of Excellence in Plasma Science and Electromagnetic Waves, Walailak University (Fig. 1). The MI output powers were adjusted to 740, 780, and 810 W in order to meet the drying temperature of 60ºC, 70ºC, and 80ºC as done by a traditional HA drier.”??used conditions??tray dryer air speed??

Ans: It was changed to “Two drying techniques including HA and MI were used to prepare TCP. The fresh samples (1,000 g) were uniformly spread in the tray with the thickness of 0.5 cm and subjected to dry using a traditional HA drier or an MI drier. The HA drier was operated in a DT 20S tray drier (4,500 W, Owner Foods Machinery Co., Ltd., Bangkok, Thailand) at 60 �C, 70 �C, and 80 �C with the circulation speed of 1 m/s. It has been reported that the maximum temperature used for drying the whole stink bean (Parkia speciosa) seed was at 70�C using the same HA (tray drier) [5]. So, in this study, the drying temperatures were varied on the basis of 70�10 �C which were 60 �C, 70 �C, and 80 �C for TCP. The MI drier used in this study was developed by the Center of Excellence in Plasma Science and Electromagnetic Waves, Walailak University (Fig. 1). The microwave output powers were adjusted to 740 W, 780 W, and 810 W with a fixed power of infrared heating (500 W) in order to meet the drying temperature of 60 �C, 70 �C, and 80 �C as done by a traditional HA drier.”

Line 115- “The fresh samples were uniformly spread in the tray with the thickness of 0.5 cm and subjected to dry using a traditional HA drier (tray drier) or an MI drier.”??amount used??

Ans: The fresh samples (1,000 g) were uniformly spread in the tray with the thickness of 0.5 cm and subjected to dry using a traditional HA drier or an MI drier.

Line 123- “The obtained TCP was packed in an aluminum foil laminated bag to prevent moisture adsorption and kept in an auto desiccator at room temperature for 24 h.”??packaging dimensions??

Ans: It was “10 cm � 15 cm”. It was added in the text already. “The obtained TCP was packed in an aluminum foil laminated bag (10 cm � 15 cm) to prevent moisture adsorption and kept in an auto desiccator at room temperature for 24 h.”

Line 167- “Colorimetric values of the samples were measured using a Hunterlab colorimeter (Hunter Assoc. Laboratory; VA, USA). The L*, a*, and b* values were recorded.”???illuminant used??ºobserver??calibration??

Ans: It was detailed “Colorimetric values of the samples were measured using a Hunterlab colorimeter (Hunter Assoc. Laboratory; VA, USA) with 10 standard observers and illuminant D65. The instrument was calibrated to a white and black standard. The L* (lightness), a* (redness/greenness), and b* (yellowness/blueness) values were recorded.”

Line 184- “Wettability”??or solubility in water??

Ans: It was a wettability, according to the references (no. 26 and 27).

Results and discussion

This section has lack of depth and must be improved.

Ans: This section was intensively revised. 

Line 251- “From the results, TCP contributed not only for taste and flavor enhancers, but also played a part in an extra-source of nutrients.”?? Aroma???flavor??measured??

Particle size??

Ans: The aroma, flavor, and particle size were not determined in this study. Based on the nutrient compositions in the proximate analysis, it was changed to “From the results, TCP can be classified as a nutritive flavoring agent.”

Figure 1- Please define each component.

Ans: Fig 1 was revised and labelled.

Figure 4- Please add wavenumber in each peak.

Ans: Wavenumbers were added in each peak.

Conclusion

Line 451- “Results suggested that TCP prepared by MI method showed the superior physico-chemical characteristics to the traditional HA.”???superior??in which results??

Ans: Conclusion was revised intensively. (see Conclusion).

References

37 references have more than 5 years. Please update your list of references.

Ans: The references were updated.

Reviewer #4: In the present study titled “Physico-chemical aspects of Thai fermented fish viscera, Tai-Pla, curry powder processed by hot air drying and hybrid microwave-infrared drying” is presented in detailed a well elaborated research which evaluated the properties of Thai fermented fish viscera, Tai-Pla, curry powder (TCP) dryed on different approaches: hot air drying (HA) and hybrid microwave-infrared drying (MI). The properties of both fresh and dried powders were evaluated by proximate composition, aw, total phenolic content, DPPH radical scavenging activity, TBARS, FT-IR spectroscopy and physical characteristics (colour, browning intensity, bulk density, wetability).

The research led to the identification of the best drying procedure of the fresh pasta with the desired properties, namely the hybrid microwave-infrared drying (MI).

In general the data are strong, and convincingly shows that the hybrid microwave-infrared drying (MI) approach could be used as a drying procedure to obtain good quality food products. The manuscript is well written, concise and the appropriate analyses are performed.

Overall, this is a well performed study that I consider that is important and represent a new strategy to conveniently obtain Thai fermented fish viscera, Tai-Pla, curry powder (TCP) with good quality characteristics.

The authors need to address the below comments to strengthen the quality of the manuscript:

1. Please insert the characterization methods used in the present study in the Abstract (e.g. FT-IR, DPPH assay).

Ans: The DPPH and FTIR results were added in the Abstract.

2. Please replace the phrases from line 37 and 455: „Therefore, MI was a promising drying technique to reduce the drying time and improve the overall quality of TCP.” By „Therefore, MI is a promising drying technique to reduce the drying time and improve the overall quality of TCP.”

Ans: Done.

3. In the preparation method of the pasta (in Materials and methods) please include the mass percentage of the main ingredients used to obtain the product.

Ans: The mass percentage of the main ingredients used for production of Tai-Pla curry was given.

Reviewer #5: In this manuscript, the authors compared the effect of hot air drying (HA) and hybrid microwave-infrared drying (MI) on physico-chemical characteristics of Thai fermented fish viscera, Tai-Pla, curry powder (TCP). The data in this article is not solid and well analyzed. Besides, the article doesn't present the application superiority of MI thoroughly, in other words doesn't fit with the average originality found of PLOS ONE. I recommend to not publish the article.

Ans: All data obtained scientifically from the experiments. The research justification, background, experimental design, and methodology were optimally stated. This research was the first report on the production of TCP using MI in comparison with HA which can guarantee the originality of the work.

Major comments

The data succeed to show the different effect of HA and MI, but the experiments are too superficial, as well the test of antioxidant activities. According to the introduction, the application of TCP mainly gives the Thai food a special flavor and aroma, which should be concerned in this article because the HA and MI processing would change the flavor of TCP. In the overall the text is well written but is superficial and is out of the journal standards.

Ans: Thank you very much for your invaluable suggestion. Regarding the flavor and aroma, we will do the sensory analysis in the future. However, from our own observation, the sensory aspects (odor, flavor, taste) of the TCP were remained unchanged. The recommendation regarding this issue was added in the Conclusion. 

Detailed comments

1. The authors developed a hybrid microwave-infrared drier but the schematic diagram (Fig. 1) didn’t show how the hybrid microwave-infrared drier works.

Ans: Fig 1 was revised and labelled.

2. SEM is recommended to show the influence of these physical processes on tissues of TCP during drying.

Ans: Thank you for your invaluable suggestion. Unfortunately, the SEM was not measured in this study. We will do it in the future work. Basically, without SEM, the methods used in this study can characterize the physico-chemical properties of the TCP.

3. Line 36: better change “between … to …” to “from … to …”.

Ans: Done.

4. Line 61: “a” should be “its” or deleted.

Ans: Deleted.

5. Line 76-77: Please check this sentence and revise.

Ans: It was changed to “MW is based on dielectric heating by electromagnetic waves and it has several advantages [5].”

6. Line 163: FTIR spectra of fresh samples can’t be found in Figure 4.

Ans: It was changed to “FTIR spectra (400-4000 cm-1 with the resolution of 4 cm-1 at the average of 16 scans) of the TCP were obtained using…….”

7. Line 251-252: Please check this sentence and revise.

Ans: It was changed to “From the results, TCP can be classified as a nutritive flavoring agent.”

8. Line 271: Please check this sentence and revise.

Ans: It was changed to “This was probably due to the evaporation of moisture, leading to the increased TPC concentration in the dried samples.”

9. Line 380-381: Please check this sentence and revise.

Ans: It was changed to “With increasing temperature in HA, L* and b* values decreased but a* value remained constant (Table 2).”

10. Line 411-412: Please check this sentence and revise.

Ans: It was changed to “In addition, MI quickly produced heat and effectively removed water at surface. This can shorten the drying time and hence reduce the degree of browning reactions.”

11. Line 421: “P<0.05” is wrong.

Ans: It was changed to “p>0.05”.

12. Please replace “governed” with another word in line 30, line 307, and line 441.

Ans: It was changed to “influenced”, “affected”, and “influenced”, respectively.

13. Line 51: “cuisin” should be “cuisine”.

Ans: Done.

14. Please add a comma before the second subject in line 215-217, line 217-218, and line 402-403.

Ans: Done.

15. Please mark “a, b, c” in the Figure 3 according to “Fig. 3a, Fig. 3b, Fig. 3c” in line 302-313.

Ans: Done.

Reviewer #6: In this manuscript, Choopan et al. compared the hot air (HA) and hybrid microwave-infrared (MI) drying ways on the Physico-chemical characteristics of TCP. They have demonstrated that MI drying with a condition of 810 W for 40 min effectively reduced the drying time by five-fold. They also showed that MI-dried TCP had the lowest browning index, the highest lightness, higher phenolic content, and lower TBARS, indicating improved overall quality. The manuscript has convincing data to support their conclusion.

The concern I have is whether MI drying affects the flavor of TCP. Is it possible to evaluate the flavor?

Ans: Thank you for your invaluable suggestion. This study aimed to explore the physico-chemical aspects of the TCP. We will do the sensory analysis in the future. However, from our own observation, the sensory aspects (odor, flavor, taste) of the TCP were remained unchanged. The recommendation regarding this issue was added in the Conclusion.

Reviewer #7: The paper presents an application of Physico-chemical aspects for Thai fermented fish viscera, Tai-Pla, curry powder processed. It is a topic of interest to the researchers in the related areas but this paper needs improvement before acceptance for publication. My detailed comments are as follows:

1. The sample source and size are both important for this paper, and please provide more details about the sample information in the section Tai-Pla curry paste preparation.

Ans: The mass percentage of the ingredients used for production of Tai-Pla curry and the sample information were given. 

2. Instrument model and manufacturer used for the drying process are not provided in the section Drying experiment. Please provide more details about these information.

Ans: Instrument model and manufacturers were given.

3. In the sections “Total phenolic content (TPC) and DPPH radical scavenging activity”, “Thiobarbituric acid reactive substances (TBARS)”, and “Fourier transform infrared (FTIR) spectroscopy”, only 20 g (i.e., 10g for Total phenolic content (TPC) and DPPH radical scavenging activity and 10 g for Thiobarbituric acid reactive substances (TBARS)) is not enough for us to consider the robustness of your chemical measurements. You must provide more samples to ensure it.

Ans: The sample sizes were used following the published standard methods from references. Also, the analyses were run in triplicate. So, measurements were reliable and repeatable. 

4. Please provide more details in the section Fourier transform infrared (FTIR) spectroscopy and Color.

Ans: Done.

5. In the section “Statistical analysis”, please provide the sample size and the sample category for the data analysis.

Ans: Done. (See Statistical analysis section).

6. The increase of TPC from the fresh sample to the drying sample is not reasonable in the Figure 3. Please show us more reason or information about it .

Ans: The possible reasons were given. 

“A slight increment in TPC was observed in TCP compared to the fresh sample (Fig. 3a). This was probably due to the evaporation of moisture, leading to the increased TPC concentration in the dried samples.”

“The increment of phenolic compounds after drying could involve in the cell wall breakdown and disruption by heating effects and consequently eases the release and extractability of bound phenolic compounds [39]. Generally, plant phenolic compounds can be classified into two major forms, free and bound [41].”

Reviewer #8: The manuscript entitled “Physico-chemical aspects of Thai fermented fish viscera, Tai-Pla, curry powder processed by hot air drying and hybrid microwave-infrared drying” investigated the effect of hot air drying (HA) and hybrid microwave-infrared drying (MI) on physico-chemical characteristics of Thai fermented fish viscera, Tai-Pla, curry powder (TCP). The present manuscript requires major revision before considering for the acceptance.

1. Add references in the line 54-64

Ans: References were added.

2. Mention the quantity of each ingredients taken for the preparation of Tai-Pla curry paste (Line 102-104)

Ans: Quantity of each ingredients were mentioned.

3. The authors mentioned that the fresh curry paste was pasteurized at 90oC for 10 min. There is any standard pre-optimized protocol/references available?

Ans: Reference was added. 

4. In case of MI how output power level calculation was done? And what basis the power level of 740, 780, and 810 W was chosen?

Ans: The microwave output powers were adjusted to 740 W, 780 W, and 810 W with a fixed power of infrared heating (500 W) in order to meet the drying temperature of 60 �C, 70 �C, and 80 �C as done by a traditional HA drier.

5. Elaborate the standard methodology (AOAC) followed to determine moisture, protein, fat, ash, and carbohydrate

Ans: The standard AOAC’s methods [17] were used for proximate composition analysis including moisture (AOAC method number 950.46), crude protein (AOAC method number 928.08), fat (AOAC method number 963.15), ash (AOAC method number 920.153), fiber (AOAC method number 962.09), and carbohydrate. Carbohydrate content was estimated by difference (1).

Carbohydrate content (%)=100-[moisture+protein+fat+ash+fiber] (1)

6. In line 168, add the description for color value L*, a* and b*

Ans: It was changed to “The L* (lightness), a* (redness/greenness), and b* (yellowness/blueness) values were recorded.”

7. In line 207 author have mentioned drying time 70oC and 80 oC had the similar drying time of 120 min. Why so? If there is difference in 10 oC could achieved at 120 min?

Ans: We apologized for the mistake. From the drying curve, the moisture content of sample dried using HA at 80 �C reached the target moisture content at 100 min. So, the drying time of 80 �C was corrected to 100 min and the Results and discussion was revised accordingly. 

“TCP dried with HA at 70 �C and 80 �C had the drying time of 120 and 100 min, respectively (Fig. 2a), which was shorter than that did at 60°C for 1.75 and 2.10 folds, respectively.”

“However, HA at 70 �C and 80 �C showed a negligible different in drying rate (k � 0.3436 vs 0.3907) (Fig.2a).”

8. What would be the final temperature achieved at 740W/70 min, 780W/45 min and 810W/40 min?

Ans: We stated earlier that the final temperature achieved at 740W/70 min, 780W/45 min and 810W/40 min was 60, 70, and 80�C. “The microwave output powers were adjusted to 740 W, 780 W, and 810 W with a fixed power of infrared heating (500 W) in order to meet the drying temperature of 60 �C, 70 �C, and 80 �C as done by a traditional HA drier.”

9. Rewrite the whole section in results part -Free radical scavenging activity, TBARS- precisely with the obtained values and compare with other studies

Ans: These parts were revised.

10. Explain the effect of drying on functional group in FTIR-Rewrite the paragraph

Ans: The FTIR section was rewritten.

11. Revise the conclusion part as per the obtained results

Ans: The Conclusion was revised intensively.

12. Need to add statistical design or experimental design

Ans: Done. (see Statistical analysis section)

13. Fig 1. label the each parts

Ans: Fig 1 was revised and labelled. 

14. Fig 4 & 5 labelling is inappropriate

Ans: Fig 4 and 5 were revised.

---

## [Decision Letter · Decision Letter 1]

15 Jun 2021

Physico-chemical aspects of Thai fermented fish viscera, Tai-Pla, curry powder processed by hot air drying and hybrid microwave-infrared drying

PONE-D-21-10233R1

Dear Dr. Panpipat,

We’re pleased to inform you that your manuscript has been judged scientifically suitable for publication and will be formally accepted for publication once it meets all outstanding technical requirements.

Kind regards,

C. Anandharamakrishnan

Academic Editor

PLOS ONE

Reviewers' comments:

Reviewer's Responses to Questions

**Comments to the Author**

1. If the authors have adequately addressed your comments raised in a previous round of review and you feel that this manuscript is now acceptable for publication, you may indicate that here to bypass the “Comments to the Author” section, enter your conflict of interest statement in the “Confidential to Editor” section, and submit your "Accept" recommendation.

Reviewer #1: All comments have been addressed

Reviewer #3: All comments have been addressed

Reviewer #4: All comments have been addressed

Reviewer #6: All comments have been addressed

Reviewer #7: All comments have been addressed

Reviewer #8: All comments have been addressed

2. Is the manuscript technically sound, and do the data support the conclusions?

Reviewer #1: Yes

Reviewer #3: Yes

Reviewer #4: Yes

Reviewer #6: Yes

Reviewer #7: Yes

Reviewer #8: Yes

3. Has the statistical analysis been performed appropriately and rigorously? 

Reviewer #1: Yes

Reviewer #3: Yes

Reviewer #4: Yes

Reviewer #6: Yes

Reviewer #7: Yes

Reviewer #8: Yes

4. Have the authors made all data underlying the findings in their manuscript fully available?

Reviewer #1: Yes

Reviewer #3: Yes

Reviewer #4: Yes

Reviewer #6: Yes

Reviewer #7: Yes

Reviewer #8: Yes

5. Is the manuscript presented in an intelligible fashion and written in standard English?

Reviewer #1: Yes

Reviewer #3: Yes

Reviewer #4: Yes

Reviewer #6: Yes

Reviewer #7: Yes

Reviewer #8: Yes

6. Review Comments to the Author

Reviewer #1: (No Response)

Reviewer #3: (No Response)

Reviewer #4: The authors addressed all the comments to strengthen the quality of the manuscript: the manuscript is now well written and well technically sound. The data support the discussions and the conclusions, newer references have been added.

Reviewer #6: In the revised manuscript, the authors addressed all my questions. I have no more concerns and suggest accepting the paper for publication.

Reviewer #7: (No Response)

Reviewer #8: (No Response)

7. PLOS authors have the option to publish the peer review history of their article (what does this mean?). If published, this will include your full peer review and any attached files.

Reviewer #1: No

Reviewer #3: No

Reviewer #4: No

Reviewer #6: No

Reviewer #7: No

Reviewer #8: **Yes: **VENKATACHALAPATHY NATARAJAN

---

## [Editor Report · Acceptance letter]

18 Jun 2021

PONE-D-21-10233R1 

Physico-chemical aspects of Thai fermented fish viscera, *Tai-Pla*, curry powder processed by hot air drying and hybrid microwave-infrared drying 

Dear Dr. Panpipat:

I'm pleased to inform you that your manuscript has been deemed suitable for publication in PLOS ONE. Congratulations! Your manuscript is now with our production department. 

Kind regards, 

on behalf of

Dr. C. Anandharamakrishnan 

Academic Editor

PLOS ONE